# The Proteome Profile of Olfactory Ecto-Mesenchymal Stem Cells-Derived from Patients with Familial Alzheimer’s Disease Reveals New Insights for AD Study

**DOI:** 10.3390/ijms241612606

**Published:** 2023-08-09

**Authors:** Lory J. Rochín-Hernández, Miguel A. Jiménez-Acosta, Lorena Ramírez-Reyes, María del Pilar Figueroa-Corona, Víctor J. Sánchez-González, Maribel Orozco-Barajas, Marco A. Meraz-Ríos

**Affiliations:** 1Departamento de Biomedicina Molecular, Centro de Investigación y de Estudios Avanzados del Instituto Politécnico Nacional, Instituto Politécnico Nacional 2508, Ciudad de México 07360, Mexico; lory.rochinh@cinvestav.mx (L.J.R.-H.); miguel.jimenez@cinvestav.mx (M.A.J.-A.); mpfigueroa@cinvestav.mx (M.d.P.F.-C.); 2Unidad de Genómica, Proteómica y Metabolómica, Laboratorio Nacional de Servicios Experimentales (LaNSE), Centro de Investigación y de Estudios Avanzados, Ciudad de México 07360, Mexico; lramirez@cinvestav.mx; 3Centro Universitario de Los Altos, Universidad de Guadalajara, Tepatitlán de Morelos 47620, Mexico; victor.sanchez@academicos.udg.mx (V.J.S.-G.); maribel.orozco@academicos.udg.mx (M.O.-B.)

**Keywords:** proteome, Alzheimer’s disease, Familial Alzheimer’s disease, PSEN1, A431E, mesenchymal stem cells, proteostasis, olfactory, neurodegeneration, FAD

## Abstract

Alzheimer’s disease (AD), the most common neurodegenerative disease and the first cause of dementia worldwide, has no effective treatment, and its pathological mechanisms are not yet fully understood. We conducted this study to explore the proteomic differences associated with Familial Alzheimer’s Disease (FAD) in olfactory ecto-mesenchymal stem cells (MSCs) derived from PSEN1 (A431E) mutation carriers compared with healthy donors paired by age and gender through two label-free liquid chromatography-mass spectrometry approaches. The first analysis compared carrier 1 (patient with symptoms, P1) and its control (healthy donor, C1), and the second compared carrier 2 (patient with pre-symptoms, P2) with its respective control cells (C2) to evaluate whether the protein alterations presented in the symptomatic carrier were also present in the pre-symptom stages. Finally, we analyzed the differentially expressed proteins (DEPs) for biological and functional enrichment. These proteins showed impaired expression in a stage-dependent manner and are involved in energy metabolism, vesicle transport, actin cytoskeleton, cell proliferation, and proteostasis pathways, in line with previous AD reports. Our study is the first to conduct a proteomic analysis of MSCs from the Jalisco FAD patients in two stages of the disease (symptomatic and presymptomatic), showing these cells as a new and excellent in vitro model for future AD studies.

## 1. Introduction

Alzheimer’s (AD) is the most common neurodegenerative disease, the first cause of dementia, and one of the top seven causes of death globally. Due to the accelerated epidemiological growth of aging people, cases will triple over the next 30 years [1,2]. Currently, there are neither curative treatments nor adequate diagnostic tools. AD is characterized by the formation of extracellular amyloid β (Aβ) aggregates derived from proteolytic cleavage of the Amyloid Precursor Protein (APP), known as amyloid plaques, and by the intraneural neurofibrillary tangles (NFTs) of hyperphosphorylated tau in the entorhinal cortex, hippocampus (Hp), limbic system, and temporal cortex, principally [3]. These pathological changes start at least 10–20 years before the first symptoms manifest as progressive memory loss and decline in cognitive functions, leading to dementia [3,4].

AD can be classified into Late-Onset Alzheimer’s disease (LOAD) or Sporadic Alzheimer’s disease (SAD) when it occurs in people over 65 years old and is of unknown etiology, comprising 95% of all cases, and Early Onset Alzheimer’s disease (EOAD) if it occurs in individuals under 65 years of age [5]. Approximately 80% of EOAD relates to an autosomal dominant defect in three principal genes: Presenilin-1 (PSEN1), Presenilin-2 (PSEN2), and the APP. This inherited subgroup of AD is called Familial Alzheimer’s Disease (FAD) [6]. Due to histopathological and clinical similarities with SAD, FAD has been used as a model for AD studies. However, over 20% of FAD patients develop atypical clinical symptoms such as spastic paraparesis [7], dysarthria, schizophrenia, Parkinsonism, and depression [8,9]. Within FAD, it is worth noting that a particular mutation has been reported in Mexico, named Jalisco mutation A431E of the PSEN1. Almost half of these patients present with spastic paraparesis, language impairments, and psychiatric and motor manifestations. Due to the genetic characteristics of FAD and its complete penetrance, it is crucial to study the early stages of the disease and develop new therapies [10,11].

To accomplish this, label-free UPLC-HDMS^E^ quantitative proteomics (Ultra performance liquid chromatography coupled with an ion mobility time-of-flight high-definition/high-resolution mass spectrometer), which uses proteome analysis with the highest coverage to look at changes in the proteomics profile, can be used to find novel pathways, molecular mechanisms, and therapeutic targets that may significantly advance our understanding of the disease [12]. Recently, many proteomics studies have screened dysregulated proteins in various biological samples derived from patients with AD [13] like some postmortem brain tissues [14,15,16,17], cerebrospinal fluid (CSF) [18,19], blood (serum/plasma) [20,21] and even saliva [22]. Unfortunately, these samples have abundant proteins that can hide the identification of essential proteins or are not directly related to the brain’s disease. The most recent research models, such as patient-derived mesenchymal cells, aid in understanding and tracking the physiopathology of the illness, discovering novel biomarkers, and creating novel therapeutics and diagnostics for the prevention or treatment of AD [23,24]. 

A potential source of neural stem cells is the olfactory mucosa, where neurogenesis is necessary to replace the olfactory neurons. Olfactory ecto-mesenchymal stem cells (MSCs) have recently been discovered [25,26,27] and they possess unique properties compared to mesenchymal stem cells isolated from other tissue sources. First, these cells show a self-renewal and a clonal and neurosphere formation capacity [27,28,29]. Due to their ectoderm origin, they have a higher neurogenic potential than mesodermal lineage. However, they have been differentiated into non-neural lineage in vitro and in vivo [28,30,31,32,33] and thus have been proposed for regenerative therapy for multiple diseases [34,35,36,37]. These cells have been obtained in postmortem tissues and via invasive methods such as mucosal and epithelial biopsies. They can be obtained from the olfactory nasal niche using non-invasive techniques [25]. Additionally, these cells are crucial to the disease because most AD patients also have hyposmia years before symptoms appear [38,39].

In the early stages of the disease, studies have shown the existence of Aβ and tau protein aggregates in the olfactory pathway, olfactory bulb (OB), and neuroepithelium [40,41], as well as altered neurogenesis with lower viability of neurons in comparison to controls [42]. These considerations suggest that olfactory MSCs may offer a novel and reliable study model for AD comprehension. This study assesses presymptomatic and symptomatic conditions stages in MSCs derived from human carriers of the PSEN1 (A431E) mutation. This quantitative proteome study in question is the first of its kind. MSCs from mutation carriers and MSCs from healthy donors were compared using two label-free proteomic approaches to find variations in their overall protein expression connected to FAD and better comprehend the molecular mechanisms underlying the development of the disease.

## 2. Results

Two healthy adult donors and two PSEN1 (A431E) mutation carriers, whose clinical histories, neurological and cognitive information were gathered and described by Santos-Mandujano [43] (see Section 4) were used to obtain and employ olfactory MSCs at pass 9. Siblings who carry the mutated gene have a three-generational history of lower limb paraparesis that has progressed. Appendix A depicts the family’s pedigree. Cells from symptomatic (P1) and presymptomatic (P2) patients were employed in this study.

We use label-free quantitative proteomic analysis to assess the DEPs between presymptomatic and symptomatic MSCs and their control. Figure 1 depicts the experimental process used in proteomics investigations, thoroughly described in the procedures. We gathered and characterized olfactory MSCs using flow cytometry and collected protein data using LC-MS/MS on a Synapt G2-Si in MS^E^. The raw mass spectrometry data were analyzed with Progenesis QI software(QIP; version 3.03) and searched in WebGestalt, UniProt, Panther, Reactome, STRING, and GeneCards for biological and functional analysis.

### 2.1. Protein Expression in Symptomatic and Presymptomatic PSEN1(A431E) Carriers (P1 and P2) and Controls (C1 and C2)

The first patient with the PSEN1(A431E) mutation is symptomatic (P1). He is a 54-year-old man with a 7-year history of lower extremity motor impairment, minor cognitive impairment, typical upper motor neuron disease symptoms, and anosmia. The second carrier is at the presymptomatic stage (P2). P2 is a 44-year-old lady with normal cognitive function, little leg weariness, generalized hyperreflexia, and hyposmia. The presymptomatic control (C2) and the symptomatic control (C1) are represented by the healthy 42-year-old woman and the 55-year-old man, respectively.

#### 2.1.1. Isolation and Characterization of Olfactory MSCs

Nasal exfoliates from research participants displayed varied cellular morphologies in the initial passes. However, a uniform population of MSCs emerged following the five passes [25] (Appendix A). Cells from nine passes of flow cytometry were used to confirm the expression of specific markers for mesenchymal stem cells (CD105, CD90, and CD73) and the absence of hematological markers and differentiated cells (CD34, CD45, CD14, CD19, and CD166) [44]. Figure 2 displays histograms contrasting P1 and P2 with C1 and C2, respectively. Appendix A shows the extensive histograms by marker and sample. 

#### 2.1.2. Label-Free UPLC-HDMSE Analysis for P1 vs. C1

We conducted a quantitative label-free proteome analysis after confirming that the isolated cells are homogenous MSCs to uncover the changed protein expression profile in MSCs from the presymptomatic carrier (P1 and P2) in comparison to the MSCs from its controls (C1 and C2), respectively, Figure 3.

#### 2.1.3. Peptide and Protein Reliability and Confidence

At the peptide and protein levels, data quality control criteria were used. Ion mobility mass spectrometry (IM-MS) was used in our peptide analysis to take signals with [M + 2H]2+ or more charges into account. There were 257,971 peptides detected in the symptomatic patient and 185,976 peptides in the presymptomatic patient; accuracy was 74.72% and 83.15%, respectively, with a maximum of 5 ppm (Figure 3A,E). Based on peptide-matched type, peptides were categorized (Figure 3C,G). The peptides with the highest quality (PepFrag1) in P1 were 75.9% and 76.1% in P2, those with lower reliability (PepFrag2) were 3.7% and 2.7%, missing cleavage peptides were 13.7% and 15.8%, and variable modification peptides were 6.1% and 5.3%, respectively. Both analyses showed that the peptide fragments fragmented at the ion source were less than 1%. The PepFrag1 did not exceed an error of ±10 ppm, consistent with the full range of m/z 400–1600 (Figure 3B,F). These results show proper mass spectrometer calibration, good quality of peptides, and efficient enzymatic digestion.

At the protein level, 4750 proteins were identified in P1 and 3971 in P2, averaging 5.7 and 5.67 peptides per protein. The 2870 and 2361 proteins were not quantified, and 330 and 246 were reversed sequences. The pattern of the significant values in P1 vs. P2 was maintained. Finally, 1553 proteins in P1 and 1277 in P2 were tested, with an average of 10.34 and 9.69 peptides per protein and a dynamic range of 6.5 and 6 orders of magnitude, respectively. These results revealed that the normalization of injection had been carried out correctly, with both conditions being highly sensitively comparable (Figure 3D,H), showing that our analysis has proper reliability and confidence at the peptide or protein level based on Figures of Merit (FOMs) for label-free proteomic studies by Souza et al. [12].

Additionally, we found three unique proteins for C1, twelve for C2, and eleven and twelve for P1 and P2, respectively (Table 1). All of these findings are summarized in Appendix A.

### 2.2. Data Filtering and Differentially Expressed Proteins in P1 and P2 MSCs

We only considered proteins with more than two identified peptides per protein and at least one unique peptide, an ANOVA *p*-value ≤ 0.05, and a coefficient of variation (CV) ≤ 0.3. We removed the reversed proteins that were false positives and the proteins in each condition in one of the three duplicates. Fold change (FC) levels of downregulated and upregulated proteins, respectively, are 0.6 and >1.5 in the differential expression. With these restrictive filters, 703 and 517 highly reliable quantified proteins remained in P1/C1 and P2/C2, respectively. Of these, 359 and 195 were differentially expressed proteins (DEPs), respectively. For P1/C1, 118 proteins were upregulated, and 241 were downregulated. For P2/C2, 75 were upregulated, and 120 were downregulated, scattered in volcano plots (Figure 4A,C). Contrary to P2/C2, which had less than 50% of the proteome modified, P1 MSCs’ proteome was altered by more than 50% compared to their C1 controls (Figure 4B,D). The top 20 proteins down- and upregulated in MSCs from P1 to C1 are shown in Table 2 and Table 3, respectively. P1 had a false discovery rate (FDR) of 2.8%, C1 had an FDR of 2.5%, P2’s FDR was 1.72%, and FDR of C2 was 1.75%. Appendix A summarize all differentially expressed proteins in both carriers.

HBB (hemoglobin subunit beta) was the most downregulated protein with −189.5 FC, followed by retroelement silencing factor 1 protein (*KIAA1551)* with −52 FC. The most over-regulated proteins with these stringent filters were Guanylyl cyclase C and PRP4B, with 34.9 and 25.9 FC, respectively. For P2, the two genes that were most negatively regulated were *SCIN* and *EEF1E1-BLOC1S5*, with −14.9 and −9.7, respectively, and the two genes that were most positively regulated were *DNM1L* and *COPS8*, with 94.5 and 35 of FC, respectively. 

#### Functional and Biological Analysis of DEPs in P1/C1 and P2/C2 Analysis

We performed a functional analysis of proteins with differential expression. GO and KEGG enrichment analyses were carried out using WebGestalt. Appendix A display these outcomes and all functional enrichment analyses performed on WEBGESTALT and Gene Analytics. In contrast to P2/C2 biological processes, which are predominantly linked to leukocyte immunity, degranulation, and exocytosis, DEPs of P1/C1 are involved in protein folding and the endoplasmic reticulum stress response. (Figure 5A,D). In P1/C1 and P2/C2, the cell cortex and focal adhesions in both mutation carriers and the cell junction substrate were the critical cellular components of DEPs (Figure 5B,E). Similar to this, both carriers had the primary altered molecular activities of binding to cadherins and binding to cell adhesion molecules, whereas P1/C1 had changed cytoskeleton-binding proteins and P2/C2 had altered proteins binding to actin filaments (Figure 5C,F).

The analysis of KEGG enrichment pathways with DEPs (Table 4 and Table 5, and Figure 6) revealed enrichment in pathways involved in energy metabolism, including the pentose phosphate pathway, amino acid biosynthesis, glutathione metabolism, carbon metabolism, and citrate cycle, as well as focal adhesion pathways and actin cytoskeleton regulation, even though the altered genes were different between the P1 and P2 mutation carriers. However, distinct pathways are enriched for each carrier, including phagosome or endocytosis pathways (Table 5) in P2 and neurodegenerative disease (NDD) pathways, such as Huntington’s disease in P1 (Table 4).

We also conducted pathway enrichment studies using WebGestalt with the PANTHER and REACTOME databases (Appendix A). The sole pathway presented across all databases is the pentose phosphate route, an essential step in glucose metabolism that produces ribose-5-phosphate and NADPH. These compounds are crucial for maintaining redox equilibrium and nucleotide biosynthesis. STRING was used to predict and show protein–protein interactions (PPIs) between DEPs in MSCs from the symptomatic carrier compared to its control using the highest confidence score (0.9). According to KEGG-enriched pathways, proteins were highlighted and then categorized into clusters based on the overall biological processes they are engaged in. The leading group, in the middle of the network, with the highest number of proteins and interactions, are proteins related to protein processing in the ER (red nodes), transport and degradation mechanisms (pink cluster), and focal adhesion (purple cluster). The biological processes were energy metabolism pathways (green group), ECM–receptor interaction, and adhesion (purple cluster). We also discovered proteins (yellow group) involved in Interferon signaling (Figure 7A). However, because there were more DEPs in P1 MSCs than in the P2 carrier, the P2 MSC interactome was less complex (Figure 7B). The Interferon signaling pathways and the energy metabolism were also grouped. In contrast to P1 DEPs, protein and translation-related pathways were grouped, and the core cluster was connected to vesicle secretion and transport.

The pathways of the critical and common NDDs and proteinopathies, such as Alzheimer’s disease (AD), Parkinson’s disease (PD), Huntington’s disease (HD), prion disease (PrD), and Amyotrophic Lateral Sclerosis (ALS), were interestingly enriched according to KEGG and STRING data (Table 6).

We developed an individual interactome using these proteins and their direct protein connections since protein processing in the ER is the pathway with the most proteins active and a significant number of interactions in P1. The majority were chaperones, including *STIP1,* an adaptor protein that controls the ATPase cycles of other chaperones like *HSP70* and *HSP90*. *SOD1*, a protein implicated in ROS detoxification, and three prolyl hydroxylases (*P4HA1, P4HA2*, and *CRTAP*) downregulated. The only elevated proteins (Figure 8A) included the stress-related proteins *DCTN1, NUP214*, and *MAPK1,* as well as *CRYAB, SAR1A*, and *ERO1A*. The physiological decline of chaperones and change in proteostasis in older people may have something to do with the early proteostasis imbalance, aggregation of misfolded proteins, and increased vulnerability to stress responses in these patients.

As opposed to the P2 interactome (Figure 8B) of vesicle secretion and transport, where 13 proteins were elevated, and 12 proteins were downregulated, these proteins also associated with phagosome and lysosome pathways (*TUBB3, HLA-A, ITGA2, ITGB1, THBS1, CD63, and PSAP*).

### 2.3. Comparison of DEPs between PSEN(A431E) Mutation Carriers and with Other NDD and AD DEPs Previously Reported

To compare the pathways between the DEPs involved in mutation carriers, Table 6 highlights the most critical pathways in both proteomic analyses.

#### Correlation between the Proteomes of Olfactory MSCs from PSEN1(A431E) Mutant Carriers with Their Clinical Histories

Because our results include proteins involved in some signaling pathways that are essential in the biology of NDDs, we evaluated the conditions that might be involved using GeneAnalytics and WEBGESTALT with the OMIM and DISGENET databases (Appendix A). This allowed us to compare the proteins reported as differentially expressed and as biomarkers in other diseases. Using the WEBGESTALT tool, we found that Spinocerebellar Ataxia, distal limb weakness, and muscular weakness were the pathogenic factors that led to symptomatic DEPs, which coincides with the spastic paraparesis displayed by the symptomatic carrier. Bulbar palsy is a condition where the region controlling the lower motor neurons responsible for speaking, chewing, and swallowing is altered. This finding is noteworthy because Santos-Mandujano noted that the patient’s speech fluency, shortness of breath, and difficulty eating and drinking worsened [32].

Using GeneAnalytics, we obtained similar results for muscular dystrophy, ALS, FAD, with 32 proteins matched (3 DE in brain, *PI4KA, S100A4*, and *DES*; 4 in blood, *DNAH6, CEP152, CRMP1, ITGA11*; and 25 GeneCards inferred as *ADAM10, ALB, HSPA1A, HSPA5, MAPK1, NOS2, SOD1,* and *STAT1*), HD, and surprisingly hereditary spastic paraplegia with 24 matched genes, *ALDH18A1* as a causative mutation; 13 differentially expressed in skeletal muscle as *BAG2, CANX, SCARB2,* and *UBR5*; and 12 inferred genes (*ALDH18A1, CANX, DCTN1, FUS, ITPR1, KIF5B, KIF5C, MATR3, REEP5, SETX, SOD1,* and *VPS13D*). In the presymptomatic case, the analysis also revealed muscle degeneration and atrophy, particularly in the lower limbs, and abnormalities in upper motor neuron disease, in which weakness, spasticity, and hyperreflexia are classic symptoms. It is essential to mention that the presymptomatic patient presented hyperreflexia. GeneAnalytics shows 30 proteins matched with FAD (4 DE in blood: *CSPG4, DNAH6, TPM1*, and *TRIO; CSF1R* as causative mutation; and 26 inferred genes) and 24 with Hereditary Spastic Paraplegia (10 DE in skeletal muscle: *ACSL3, CFL2, GOLGA4,* and *ITGB1;* 3 causative mutations: *ALD18A1, SACS,* and *WASHC5*; and 14 inferred genes). These studies imply that progenitor cells of PSEN1 (A431E) mutation carriers may be connected with the clinic in these individuals, but further investigation is necessary to confirm these findings.

## 3. Discussion

Our understanding of the genesis of AD is mainly focused on the brain and neuronal cells. However, the molecular changes in olfactory mesenchymal stem cells are still unclear. It is generally known that these cells’ epigenetic characteristics, cellular capabilities, and gene expression were preserved during in vitro culture. In order to compare the differences in protein expression between olfactory MSCs from patients with the PSEN1(A431E) mutation and healthy donors, two proteome profiles from various phases of FAD were examined. The significance of these DEPs was investigated further using bioinformatics and bibliographic techniques, including gene ontology analysis, KEGG pathway analysis, PPI network creation, and compared with other AD databases.

### 3.1. Exclusive Proteins of PSEN1(A431E) Mutation Carriers

The symptomatic carrier (P1) expresses eleven distinct proteins, principally concerned with viability, proliferation, migration, and inflammation, which may account for the carrier’s olfactory MSCs’ observed difficulties in growth and proliferation.

The proteins involved in neurogenesis, brain development, self-renewal, survival, and plasticity include *WDR62, MCPH1*, and *CDK5RAP2.* Numerous neurological disorders, including Alzheimer’s disease and microcephaly, have been linked to mutations in these proteins [45,46]. *WDR62* and *CDK5RAP2* may indirectly affect the growth of neurofibrillary tangles (NFTs) since *JNK* and *CDK5* kinases regulate tau phosphorylation [46,47].

The expression of histone H1.5, encoded by the *UTY* and *HIST1H1B* genes, may impact genes associated with the pathophysiology of Alzheimer’s disease. *H1.5* influences chromatin’s accessibility and structure, and *UTY*, a histone demethylase, is implicated in the epigenetic regulation of gene expression and cell development. After being identified in amyloid plaques, *HIST1H1B* has been found to have the ability to produce amyloid-like fibers [48,49].

*KIF5C* is a kinesin that regulates organelles’ mobility across microtubules and mitochondrial transport. In PS1(E280A) mutant carriers, *KIF5C* decreased in mitochondria but increased in synaptosomal-enriched fractions [50]. *SCUBE2* participates in SHH signaling and, when overexpressed, decreases glioma cell proliferation, migration, and invasion [51]. *PIGN* mutations cause epileptic encephalopathy and MCAHS1 (multiple congenital anomalies-hypotonia-seizures syndrome). The protein *PIGN* transports phosphatidylethanolamine (PE) from the ER to the first glycosylphosphatidylinositol (GPI) mannose. In fibroblasts from individuals with PSEN1 and PSEN2 mutations, elevated PE levels have been discovered in mitochondria, plasma membranes, and ER-associated mitochondrial membranes (MAMs). These modifications may increase γ-secretase activity and cause the production of Amyloid-β [52].

Furthermore, MCAHS1 mutations created by CRISPR/Cas9 in C. elegans result in protein aggregation and the activation of the unfolded protein response (UPR) pathway in the ER [53]. *ATP5PF*, a component of the mitochondrial ATP synthase in charge of producing ATP, has been linked to Alzheimer’s disease, neurodegeneration, and oxidative stress as early manifestations of this pathology [54,55]. The functions of *KIAA0825* and *ANKRD36C* are unknown; however, they may be connected to several activities, including inflammation [56].

The unique proteins found in the presymptomatic PSEN1(A431E) mutation carrier are primarily involved in transcriptional regulation (*RTRAF, DIP2C*, and *PROX1*), mitochondrial function (*ATP5H* and *TIMM13*), proliferation control (*CNBP* and *PROX1*), neuronal development (*PROX1* and *DIP2C*), and cellular transport (*FCHO1, DYNLRB1, BIN3*, and *CUL9*), suggesting a potential compensatory or resistance process in the cell to maintain proper cell function. Interestingly, neurological illnesses such as autism, neurodevelopmental disorders, deafness dystonia syndrome, microcephaly, Alzheimer’s disease, and several cancer types have been associated with *RTRAF, DIP2C, YARS, CNBP, BIN3*, *DYNLRB1*, and *TIMM13.* In addition, *DIP2* paralogues and spastic hemiplegia are connected [57].

Tyrosyl-tRNA synthetase (*YARS*) mutations have been linked to Charcot–Marie–Tooth disease. This neuropathy causes progressive loss of muscle tissue and touch sensitivity in different body parts [58]. In a cell model of tau-negative frontotemporal lobar degeneration (FTLD) and TDP-43 aggregation, a protein that accumulates in ALS, higher nuclear levels of YARS have also been found [59]. Type 2 myotonic dystrophy and sporadic inclusion body myositis disease (sIBM), an inflammatory muscle condition defined by the buildup of intramuscular Amyloid-β aggregates similar to those found in Alzheimer’s disease, increasing muscle weakening and atrophy [60], both of which are associated with *CNBP* [61]. These all resemble the spastic paraparesis experienced by carriers of this mutation and may be caused by proteins connected to the pathophysiology of these patients. 

In conclusion, the presence of these particular proteins in presymptomatic PSEN1(A431E) mutant carriers may point to a complex interaction of biological mechanisms that attenuates the effects of the mutation in MSCs at the early stages of the disease.

### 3.2. Downregulated Proteins in PSEN1(A431E) Mutation Carriers

Most downregulated proteins in symptomatic carriers are distinct proteins that appear to be connected to changes in cell proliferation, cycle arrest, differentiation, susceptibility to damage, and ROS generation.

Unexpectedly, the α and β chains of hemoglobin, the protein that transports oxygen and carbon dioxide in cells of the erythroid lineage (FC -27.6 and -189.5, respectively), are two of the most downregulated proteins in symptomatic mutation carriers compared to the control. Hb chains, however, have been found in non-erythroid cells [62], in primary cell cultures of glial cells and neurons from the murine brain, and even in Amyloid-β deposits [63,64,65]. Although mesenchymal cells have not been shown to express Hb, two Hb chains were discovered in a proteomic analysis of the olfactory bulb in 2012 [66].

Our results support Ferrer I. et al. [67], who found that nearly all neurons in senile plaques with hyperphosphorylated tau deposits and Amyloid-β-core decreased in both hemoglobin chains. The downregulation of Hb chains indicates a change in oxygen homeostasis and reactive oxygen species detoxification.

*KIAA1551* (RESF1-retroelement silencing factor 1) is necessary to maintain the repressive gene state in undifferentiated embryonic stem cells [68]. Downregulation of *RASA2, CEP192, ZGRF1*, and *ITGA2* reduces cell migration, adhesion, and proliferation while also causing cell cycle arrest, which may have an impact on the phenotypic and the course of the disease as it is now understood to exist. The maturation of the mitotic centrosome and the formation of the bipolar spindle both depend on the protein *CEP192* (Centrosomal Protein 192). Cytosolic microtubule organizing centers (MTOCs) were in excess in *CEP192* knockdown cells [69]. Several DNA-damaging substances, including PARPi and radiation, boosted the sensitivity of *ZGRF1* null cells [70].

Integrin alpha 2 (*ITGA2*), an extracellular matrix receptor for laminin, collagen, fibronectin, and E-cadherin, plays a role in promoting cell proliferation and invasion in different types of cancer [71]. Cancer cells lost their ability to differentiate into epithelial cells and experienced a motility limitation due to the loss of the 2-integrin subunit [72,73]. With an FC of −3.8, *ITGA2* is also one of the most downregulated proteins in the presymptomatic carrier.

Interferon induces several proteins, including *MX1* (Interferon-induced GTP-binding protein Mx1), *OAS3* (2′5′-oligoadenylate synthetase 3), and *IFIT3* (Interferon-induced protein containing tetratricopeptide repeats 3), whose downregulation may prevent cell migration and proliferation. It is interesting to note that, in contrast to our findings, MxA expression has been found in senile plaques and reactive microglia in AD brains [74], and MxA polymorphisms have been linked to an increased risk of Alzheimer’s disease, faster cognitive decline, and Multiple Sclerosis [75]. Apoptosis results from the knockdown of *IFIT3* in lung epithelial cells [76].

Desmin and Dysferlin are among the top 20 most downregulated proteins that are noteworthy to mention because of their involvement in myofibrillar myopathies and muscular dystrophy, which are diseases characterized by progressive skeletal muscle weakness, atrophy, autolysis/proteolysis, and autophagy of muscle cells [77]. 

Several oncogenes are among the proteins that are downregulated in the presymptomatic carrier, which may also indicate alterations in protein trafficking, apoptosis, and proliferation. Interestingly, the symptomatic carrier was altered by four of the proteins that were most heavily downregulated in the presymptomatic carrier. *SCIN* and *LRP1B* were downregulated, but *UTY*, a particular symptomatic carrier protein, was upregulated.

*SCIN, LRP1B, CPNE3, TICRR*, and *PPP1R7* are connected to cancer cell invasion, migration, proliferation, and apoptosis. By organizing the cytoskeleton, the calcium-dependent actin filament protein, *SCIN* regulates exocytosis and vesicle transit [78]. Dental pulp stem cells (DPCs) proliferated less and moved less when *SCIN* was knocked down [79].

Since LDL receptors are ApoE receptors and control both the clearance of Aβ and the amyloidogenic processing of APP, they are connected to the pathogenesis of AD [80,81]. PSEN1(A431E) mutant carriers exhibit downregulation of the tumor suppressor gene LRP1B (low-density lipoprotein receptor-related protein 1B). *LRP1B* and APP interact, preventing APP from being internally processed and not favoring amyloidogenic processing [82]. Additionally, the complement C1q protein promotes *LRP1B* expression and protects immature and adult neurons against fibrillar and oligomeric Aβ toxicity [83]. However, our findings indicate a decreased neuroprotective impact against Aβ toxicity in MSCs.

The macromolecular tRNA synthase complex includes the auxiliary protein-coding gene *EEF1E1-BLOC1S5, AIMP3.* When it was lacking in mouse embryonic stem cells (mESCs), an accumulation of DNA damage occurred, which caused p53 to become transcriptionally active and lead to a loss of stemness and differentiation potential [84]. However, Kim et al. discovered that *AIMP3* is inhibited under hypoxia, slowing cell senescence, lowering mitochondrial respiration, and activating autophagy. Additionally, it is favorably controlled by Notch3 in human placenta amnion-derived mesenchymal stem cells (hpMSCs) [85].

Non-muscle myosin II heavy chain (*MYH10*) inhibits the Wnt/β-Catenin pathway and decreases glioma cell migration and invasion [86].

*NTRK2*, which codes for the tropomyosin receptor kinase TrkB, a neurotrophin receptor with a high affinity for BDNF, is another protein among the most downregulated that merits mentioning because it has been suggested as a potential candidate gene for AD [87].

Heat shock proteins (*HSP27, HSP90*, and *HSP70*) and mtp53 are downregulated by the knockdown of ribophorin II (*RPN2*), an essential subunit of the N-oligosaccharyl transferase (*OST*) complex, in breast cancer stem cells [88]. *RNF6* is an E3 ubiquitin ligase protein participating in IFN-antiviral response and proteasome degradation [89]. Additionally, *RNF6* is silenced, which increases GSK3 activity, decreases p-GSK3 inhibition of the Wnt/β-Catenin pathway in cancer cells, and suppresses MAPK/ERK signaling, which in turn reduces cell growth and proliferation [90,91].

### 3.3. Upregulated Proteins in PSEN1(A431E) Mutation Carriers

Many of the overexpressed proteins in symptomatic carriers are linked to cell proliferation, migration, and cell cycle progression via Notch1, β-Catenin, NF-kB, AKT, and ERK1 signaling, and pathways closely related to AD, including *PRPF4B, S100A16, PRPS2, PIK3C2B, MAPRE1*, and *SVEP1*. These proteins are also overexpressed in various cancer types. There is evidence for both the opposite effects and suppression of cell proliferation, such as the overexpression of PRP4 in hepatocarcinoma cells (HCCs), which caused cell growth retardation by inducing cell cycle arrest at the G1/S checkpoint [92], and *S100A16,* which inhibits proliferation, migration, and invasion via the JNK/p38 MAPK pathway [93].

Also, Guanylate Cyclase 2C or Heat Stable Enterotoxin Receptor (*GUCY2C* or GC-C), the most upregulated protein in the symptomatic carrier, inhibits phosphodiesterase 3 (PDE3), inducing cell cycle arrest, cytostasis, and senescence via activation of p21, PKGII, and p38 MAPK [94], inhibits proliferation by decreasing AKT signaling and β-catenin and promotes cell differentiation and migration in Colorectal Carcinoma Cells [95,96]. Consequently, the overexpression of these proteins may be a factor in the symptomatic mutation carrier’s diminished migration and proliferation.

The S100 calcium-binding protein A16 (*S100A16*) has been linked to psychiatric disorders, depression, and neurodegeneration [97]. *S100A16* promotes the Wnt/β-Catenin signaling pathway and the HRD1-induced ubiquitination and degradation of GSK3 and CK1 in cases of acute renal injury [98]. ER stress indicators like GRP78, p-IRE1, and XBP1s were upregulated in HK-2 cells after S100A16 overexpression [99].

A key player in the production of purines and pyrimidines, phosphoribosyl pyrophosphate synthetase 2 (*PRPS2*) acts as an oncogene by controlling the activities of matrix metalloproteinase 9 and the expression of E-Cadherin [100,101]. The tumor suppressor gene *TSC2* (Tuberous Sclerosis Complex subunit 2 or Tuberin) negatively controls mTORC1 signaling and is crucial for autophagy and dendritic formation [102,103]. While the frontal cortex of people with Alzheimer’s disease or Down syndrome has decreased Tuberin levels [104], other studies have found a significant upregulation of phospho-Tuberin (Thr1462) in the post-mortem frontal cortex of people with AD and PD/DLB, which is mediated by the Akt-PTEN pathway [105]. TSC2 is linked to neurological deficits like epilepsy, autism, and intellectual disabilities.

*EB1* or *MAPRE1* controls the development of spindles, the stability of chromosomes, and microtubule architecture. Recently, Hahn et al. [106] reported that Eb1, *XMAP215*, and tau cooperate to promote the polymerization of microtubules (MTs), are also involved in the maintenance of mitochondrial morphology and dynamics, interact with Bax and Bak to promote cell apoptosis via reactive oxygen species [107], form heterodimers with EB2 to promote inner mitochondrial membrane degradation [108], and can regulate autophagosome biogenesis by interacting with Beclin-1 to enhance PI3KC3 [109].

The transmembrane immunoglobulin superfamily member *DSCAML1* plays a role in several neurodevelopmental processes, including branching, migration, synaptogenesis, and dendritic self-avoidance. Dscam2 also inhibits the deposition of synaptic vesicles through an endosomal route dependent on phosphatidylinositol-3 kinase (PI3K) [110].

*NAGA* was proposed as a genetic risk for schizophrenia [104] because it encodes the lysosomal enzyme alpha-N-acetylgalactosaminidase, which is necessary for the breakdown of glycolipids and helps to maintain and regulate the dendritic spine [111].

Another interesting product among the most downregulated proteins is the endosomal protein *STAM2* (Signal Transducing Adaptor item 2), a part of the ESCRT-0 complex that regulates receptor signaling and trafficking. It plays a regulatory role in the endosomal sorting of ubiquitinated membrane proteins [112]. It is strongly expressed in stomach cancer, neurons, and olfactory epithelium [113]. In HeLa cells, STAM2b overexpression results in the expansion of early endosomes and the accumulation of ubiquitinated proteins and ligand-activated EGF receptors. This behavior has already been observed in AD [114]. The scaffold protein *SRRM2* is overexpressed in the brain and blood of people with Parkinson’s disease [115] and is necessary to establish nuclear speckles and mRNA splicing [116]. Dysfunction of *SRRM2* has been connected to neurodevelopmental disorders. Additionally, it accumulates and mislocalizes from cytosolic tau NFTs to nuclear splicing speckles in cells, mice, and the brains of AD patients [117]. *SDHA* takes part in both the tricarboxylic acid (TCA) cycle and the mitochondrial electron transport chain (ETC). Inflammation-promoting fumarate buildup and enhanced mitochondrial respiration have been linked to increased *SDHA* function [118].

Phosphoglycerate dehydrogenase (*PHGDH*) upregulated in the subventricular zone (SVZ), a neural stem/progenitor niche, may play a part in neurogenesis. Severe neurological abnormalities in *PHGDH* knockout mice include congenital microcephaly and psychomotor impairment [119,120]. *PHGDH* and the astrocytic L-serine biosynthesis pathway are diminished in the AD brain and 3xTg-AD mice, according to a recent study by Le Douce et al. [121]. In contrast, Chen et al. later quoted Le Douce and provided results that concur with our findings, showing that *PHGDH* mRNA and protein levels rise as AD pathology and symptoms worsen in 3xTg-AD and P301S tau transgenic mice (PS19) and human AD brains [122]. Interestingly, in the presymptomatic carrier data (FC-2.3), this protein is downregulated rather than increased.

One of the most increased proteins in the presymptomatic carrier is Dynamin-related protein 1 (*DNM1L* or Drp1), a GTPase essential for mitochondrial fission. Abnormal mitochondrial dynamics, including increased fragmentation and decreased fusion, have been seen in Alzheimer’s disease (AD). In AD neurons, elevated *DNM1L* activity led to mitochondrial fragmentation and dysfunction [123], while in AD mice models, inhibiting *DNM1L* prevents cell death, enhances synaptic function, and increases neuronal activity [124].

*COPS8*, or *COP9* signalosome subunit 8, controls cellular processes such as protein breakdown, DNA damage response, and cell cycle progression. In murine livers, *COP9* deficiency results in Ubiquitin Proteosome System (UPS) dysfunction and apoptosis [125]. In a mouse model of cardiac proteinopathy, the COP9 signalosome regulates the breakdown of cytosolic misfolded proteins and guards against proteotoxicity [126]. Colony-stimulating factor 1 receptor (*CSF1R*) is a transmembrane protein that aids in the differentiation and survival of microglia. It is elevated in post-mortem samples from AD patients [127,128], and selective deletion of *CSF1R* in forebrain neurons in mice accelerated excitotoxin-induced death and neurodegeneration [129].

However, other studies showed that pharmacological inhibition in APP/PS1 mice improved performance in memory and behavioral tasks and prevented synaptic degeneration [128].

*FKBP1A*, a member of the immunophilin protein family involved in protein folding and trafficking, was protective against the cytotoxicity caused by Aβ in cultured cells when overexpressed [130]. Cognitive deficits have been associated with the increased expression of other members of this family, such as FKBP51 [131].

Obg-like ATPase 1 (*OLA1*) lessens the antioxidant response, strengthening the integrated stress response and controlling protein production. *SAMHD1* is likewise elevated in the symptomatic carrier (both with FC of 1.7). Its overexpression makes cells more vulnerable to cytotoxicity from thiol-depleting and peroxide antioxidants [132]. But according to R-F Mao et al., *OLA1* binds to and stabilizes HSP70 to shield cells from heat shock [133]. Overexpression of *CPT1C* lowered cell viability, oxidative stress, and apoptosis in Aβ25-35-induced neurons and decreased APP, p-tau, and Bace-1 levels. *CPT1A* is an enzyme that transports fatty acids into the mitochondria to produce energy [134].

### 3.4. Signaling Pathways and Biological Processes Altered in PSEN1(A431E) Mutation Carriers

The term “proteostasis network” (PN) describes the interaction of a complex network of processes that allow the maintenance of the conformation, concentration, and localization of proteins for their correct function. This network comprises about 2500 genes (>10% of the total protein-coding genes) and multiple signaling pathways that control protein biogenesis, folding, trafficking, and degradation. Any disruption in Protein Quality Control (PQC), changes in glucose uptake and metabolism, oxidative stress, transport, and abnormal autophagy or degradation systems as a result of stress can result in misfolding or accumulation of proteins that interfere with various cellular functions and disrupt the entire proteostasis network, as happens in aging and the main NDDs, including AD [135], causing oxidative stress, neuroinflammation, and toxicity leading to dysfunction and cell death [136,137].

Our findings demonstrate that the primary categories that are altered in MSCs derived from carriers of the PSEN1(A431E) mutation include proteostasis network functions like energy metabolism, processing, synthesis, and folding of proteins, cytoskeleton organization, and intracellular and vesicular transport (Figure 9).

According to some studies, bioenergetic changes, including mitochondrial malfunction, changes in glycolysis and PPP, reduced ATP generation, and oxidative damage are early symptoms of AD [138,139,140].

One of the characteristics of AD is a decline in cerebral glucose absorption. The changes in glucose hypometabolism in pre-symptomatic AD are now being studied using MRI and 18F-FDG PET (Brain Fluorodeoxyglucose Positron Emission Tomography) imaging investigations [141,142]. A group of individuals from FAD families with mutations in PSEN1 who were presymptomatic had a glucose hypometabolism, according to Mosconi et al.’s findings [143].

Most DEPs participating in energy metabolism pathways were downregulated, pointing to disordered energy metabolism and a potential rise in ROS generation. Although elevated in P1, *PHGDH* and *ALDH18A1* are downregulated in P2. It is interesting to note that the gene for *ALDH18A1* (Delta-1-Pyrroline-5-Carboxylate Synthase), a condition with symptoms comparable to those of people with PSEN1(A431E) mutation, is associated with spastic paraplegias (SPG9A and SP9B) [144]. AD and Down syndrome were also linked to *ALDH18A1* mutations [145]. *PGD, MAT2A*, and *PKM* were the three proteins whose expression levels were downregulated in both mutation carriers. The pentose phosphate pathway (PPP) has three enzymes. The third enzyme is *PGD* (phosphogluconate dehydrogenase). In contrast to our results, an old work reported that G6PD and PGD activity increased in AD patients’ inferior temporal cortex [146]. However, Tiwari and Patel said that the PPP flux is decreased in AβPP-PS1 mice [147]. *MAT2A* (Methionine Adenosyl transferase 2A) is essential for the methylation of neurotransmitters, DNA, proteins, and lipids. As our results, Schrötter et al. [148] reported a lower expression of *MAT2A* in frontal cortex samples from AD patients than in controls. The knockdown of the APP protein family resulted in its downregulation at the RNA and protein levels but with a subsequent S-adenosylmethionine (SAM) elevation. To produce ATP and pyruvate, the pyruvate kinase (*PKM*) catalyzes the transfer of a phosphoryl group from phosphoenolpyruvate to ADP. *PKM2* controls the activity of γ-secretase and is necessary for cell proliferation. Contrary to our findings, Han et al. claimed that its expression is increased in brain samples from AD patients and mouse models and that its silencing led to lower levels of Aβ1-40 and Aβ1-42 and γ-secretase activity [149].

Regarding the mechanisms of degradation, there are numerous signs that AD plays a part in the endo-lysosomal, autophagic, and ubiquitin–proteasome systems, including the marked enlargement of endosomal compartments, the gradual accumulation of autophagic vacuoles (AVs), lysosome dyshomeostasis, the accumulation of ubiquitinated proteins, and changes in proteasome subunits that reduce the proteasomal activity [150].

The function and destiny of critical molecules involved in the genesis of Alzheimer’s disease (AD) depend on endocytosis. For instance, early endosomes are essential for producing Aβ peptides, which are noticeably increased within neurons in the brains of patients with early-stage AD [151,152]. These results aligned with our presymptomatic findings. The proteins Rab11, RabGDI (downregulated in our findings), phospholipase D1, syntaxins, and annexin A2 are all targets of interactions between Presenilin’s and intracellular vesicular trafficking [153]. PS1 loss also harms endocytosis and receptor-mediated transcytosis in neurons produced from iPSCs that have mutations in APP and PSEN1 [154]. Exosomes are intraluminal microvesicles that develop outside of the cell, and it has been demonstrated that exosomes can transfer pathogenic proteins like hyperphosphorylated tau, toxic amyloid beta, and other pathogens between cells that are engaged with other NDDs [155]. Defective lysosomal activity can also produce toxic Aβ and tau species oligomers, contributing to neuronal malfunction and death [152].

#### 3.4.1. Protein Processing in Endoplasmic Reticulum

Most of these events occur in the endoplasmic reticulum (ER), where most non-cytosolic proteins (secretory or membrane) are generated. A rigorous quality control system prevents around 30% of all proteins from misfolding physiologically, and chaperones, also known as heat shock proteins (HSPs), provide the first line of defense. To keep the balance between protein synthesis and degradation, they bind to unfolded proteins and keep them in a properly folded state while preventing aggregation. These proteins are grouped into various families based on their molecular weight and functions, such as cotranslational folding, refolding or degradation of misfolded proteins, and the inactivation of the Unfolded Protein Response (UPR) [156]. Sixteen molecular chaperones (*CRYAB, ERO1A, CALR, CANX, ERP29, PDIA3, PDIA4, PDIA6, BAG2, HSPA1A, HSPA5, DNAJB11, HSP90AB1, HSP90B1, HYOU1*, and *P4HB*) were identified in the clinical data of twenty-two DEPs in protein processing in the endoplasmic reticulum pathway.

Almost all chaperone protein families are altered in AD brain tissue and animal models. Many have been overexpressed due to chronic inflammation, but some reports show that HSPs are downregulated in AD [157,158,159]. The altered regulation of HSPs can increase oxidative stress or the response to the misfolded protein, potentiate the chronic overload, and functionally decrease the degradation systems that eventually lead to the development of the disease. There are conflicting reports from a few research suggesting that PSEN1 mutations in transgenic cells or animals reduce the production of chaperone proteins [160].

However, the findings that many molecular chaperones are downregulated in undifferentiated cells are noteworthy since they might play a role in the earlier clinical presentation of FAD compared to SAD.

#### 3.4.2. Regulation of Actin Cytoskeleton

Most excitatory synapses within the brain occur on small dendritic protrusions called dendritic spines. The quantity and size of dendritic spines, whose stability is provided by the actin cytoskeleton, significantly impact synaptic strength and neuronal function [161]. Actin dynamics are regulated by many molecules, particularly small GTPases like Rac and Rap. Glutamate receptors are lost from synaptic locations as a result of actin depolymerization. Similarly, spine and synapse loss are caused by disrupting the expression or activity of regulators upstream of the actin cytoskeleton, such as Rac-GEF, Rac, and Rac targets p21-activated kinases (PAK). Animal models of AD that include mutations in the susceptibility genes that cause Familial Alzheimer’s disease recapitulate the loss of the dendritic spine [162,163].

This decrease may be brought on directly by the toxicity of Aβ oligomers or indirectly by the change of several synaptic proteins in these patients [164]. In line with this, Xia et al. [165] demonstrated that AD patients and animal models of the disease have higher levels of calcineurin activation, a calcium-sensitive phosphatase. Additionally, GSK-3, a downstream effector molecule of calcineurin, has enhanced activity in response to AD-related pathologies, as demonstrated by Li et al. [166].

In addition, p21-activated kinases (PAKs), essential regulators of actin assembly and subsequent spine modulation in neurons, decrease in the hippocampus of AD patients and animal models of the disease [165]. Kalirin, a crucial regulator of spine morphogenesis and an upstream activator of PAK in dendritic spines, is also consistently under expressed in the hippocampus of AD patients regarding protein and mRNA levels. Similar to how PAKs are abnormally translocated from the cytosol to the membrane in cells from AD patients as opposed to cells from control subjects free of the disease [167]. The hippocampus of AD patients exhibits lower expression and altered location of RhoA, a small RhoGTPase that modifies cytoskeleton dynamics to modulate synaptic plasticity [168]. On the other hand, Liu et al. [169] demonstrated that the hippocampus of old mouse models and AD patients has less Drebrin protein (*DBN1*), which accumulates in dendritic spines and creates a stable pool of slowly replenishing F-actin and bundles filaments by cross-linking them together.

Aged APPPS1 mice exhibit lower levels of activated *CFL1*, an actin-binding protein that depolymerizes and cleaves actin filaments to release free G-actin monomers recruited for filament elongation and branching. Aging wild-type mice have higher levels of activated *CFL1*. This reduction is more significant in APPPS1 mice than in aging wild-type mice. Additionally, it has been noted that AD patients’ brains had lower levels of active *CFL1* [170].

#### 3.4.3. Focal Adhesion and Extracellular Matrix–Receptor Interactions

According to Bao et al., who discovered dramatically changed expression of cell adhesion pathway (CAM) genes in AD in cerebellar and temporal cortex samples from AD-affected people [171], several genome-wide association studies (GWAS) support the involvement of CAMs in AD. For example, isoform NCAM180 levels of the neural cell adhesion molecule, but not total NCAM levels, are elevated in the frontal cortex of AD patients compared to healthy subjects of the same age [172].

On the other hand, contactin-2 levels are decreased in AD patients’ temporal lobes [173] and N-cadherin levels are likewise reduced in their temporal cortex [174].

Meanwhile, single nucleotide polymorphisms (SNPs) in NCAM2 are associated as a risk factor related to AD progression in the Japanese population, according to Kimura et al. [175]. At the same time, another large GWAS encompassed more than 16,000 participants using SNPs in Contactin-5, another synaptic CAM that localizes to presynaptic membranes, were strongly linked with disease development [176], similar to what was seen with SNPs in JAM2 [177].

#### 3.4.4. Neurodegenerative Diseases Pathways

NDDs are primarily distinguished by a failure of Protein Quality Control (PQC) to remove or degrade misfolded proteins accumulating in various brain regions. NDD shares multiple pathophysiological processes that result in neuronal injury. The current study enriched KEGG and STRING data for both carriers for the major NDD pathways, including those for ALS, Parkinson’s, prion, Huntington’s, and Alzheimer’s (Table 6). These proteins play a role in protein folding, processing, and stress in the ER, disruption of the ubiquitin–proteasome system, change in calcium signaling, oxidative stress and mitochondrial dysfunction, poor autophagy, and deficiencies in microtubule transport. Only seven of these proteins were present in both PSEN1(A431E) mutant carriers, and four (*CLTA, AP2B1, DNAH6,* and *FUS*) were downregulated. In both, *RYR1* was elevated, and *TUBB3* and *MATR3* expression varied.

### 3.5. Comparison with AD Studies Previously Reported

The cytoskeleton’s structure, intracellular and vesicular transport, and energy metabolism are the primary functions that are impacted in MSCs derived from PSEN1 (A431E) mutant carriers. These processes are related to the proteostasis network and involve protein processing, synthesis, and folding.

Some AD investigations have reported proteostasis network changes, with findings similar to ours. Vesicular transport, post-translational protein modifications, trafficking, and proteostasis were among the five classes of functionally related molecular pathways linked to AD that Rosenthal et al. [178] proposed as part of their recent identification of an AD network integrating multi-omics data with the most recent genome-wide association studies (GWAS). Shokhirev and Johnson used machine learning and bioinformatic techniques in a massive multi-omic dataset of 4089 blood and brain human samples from microarray, RNA-Seq, proteomics, and miRNA-accessible data generated from AD and control participants [168]. Their findings revealed aging-related characteristics, such as cell death in the youngest patients, cellular senescence, immunological system alterations, and oxidative stress in the middle-aged group.

Johnson and colleagues conducted a detailed proteomic analysis of more than 2000 human brain tissues from 453 brains and 400 cerebrospinal fluid samples [179]. They included control and asymptomatic AD (AsymAD)were connected to disease and reflected biological processes of synapses, mitochondria, RNA binding/splicing, and astrocyte/microglial metabolism.

Additionally, Pedrero-Prieto et al. [180] created a database of CSF from AD patients from 47 different proteomic studies that were compared. Thirty-six proteins were also present in our results*: CDH6, CAMK2D, CUTA, MARCKS, SRCAP, TXN TPI1, WARS, BASP1, ALCAM, CYCS, HNRNPU, UCHL1, CHGB, HLA-B,* some related to PI3k-Akt signaling pathway (*COL1A1, COL1A2, FN1, NTRK2,* and *YWHAB*), with the lysosome (*CTSD, IGF2R,* and *PSAP*), with exocytosis, secretion and vesicle transport (*A2M, ADAM10, ALB, CANX, CTSD, FN1, FTL, GOLGB1, GSN, HBB, IGF2R, PCLO, PKM, PSAP,* and *SOD1*), and protein processing in ER (*PDIA3, PRKCSH,* and *CANX*).

Furthermore, Higginbotham et al. [181] applied an integrative proteomics approach to AD brain and CSF of healthy controls, asymptomatic, and AD patients, finding progressive disease-specific changes and dysfunctions in five main panels: immunological, metabolic, synaptic, vascular, and myelin. This suggests that disruptions of energy and redox pathways may be necessary during the preclinical stages of the disease. Another redox proteomic investigation discovered the “triangle of death” in AD brains, which comprises abnormal interactions between protein homeostasis, mTOR signaling, and energy metabolism [182].

Recent research found that compared to fibroblasts from healthy donors, skin fibroblasts from FAD patients with PS1 (M146L or A246E) mutations expressed higher levels of HSPs and autophagic-lysosomal pathway proteins [183]. Our findings demonstrated that the present work was carried out in undifferentiated olfactory MSC cells, highlighting that the pathogenic pathways were comparable to those previously documented in AD.

## 4. Materials and Methods

### 4.1. Subjects

Two healthy adult donors and two PSEN1(A431E) mutation carriers from a Mexican-Mestizo family provided the olfactory neuroepithelium. The family carriers are siblings affected with lower limb paraparesis for three generations (Appendix A). Santos-Mandujano [43] gathered data on these PSEN1(A431E) carriers, including confirmation of the mutation, clinical background information, and neurological and cognitive information. Symptomatic (P1) and presymptomatic (P2) patients’ cells were employed in this study. 

Our patient (P1) is a 54-year-old guy with progressive motor dysfunction in the lower limbs, slight cognitive impairment, typical symptoms of upper motor neuron disease, and anosmia that has been present for seven years. The presymptomatic carrier (P2) is a 44-year-old asymptomatic female with normal cognitive function, modest hyperreflexia throughout her body, and slight hyposmia. In addition, control subjects consist of two healthy donors (without the mutation) matched by age and gender: a male 54-year-old serving as the control for the symptomatic subject and a female 42-year-old serving as the control for the presymptomatic subject.

These participants provided their informed consent following the guidelines of the Bioethics Committee on Human Research (COBISH, folio # 038/2016 of the Center for Research and Advanced Studies, CINVESTAV-IPN) by signing a form authorizing the collection of their medical history and biological samples.

### 4.2. Sample Collection and Cell Culture Conditions

Nasal epithelium cells were exfoliated from the anterior region of the medial and lateral turbinate using a special toothbrush according to the methodology reported by Benítez-King et al. [25] and harvested in Dulbecco’s modified Eagle and F-12 media (DMEM/F-12 Gibco, Grand Island, NY, USA) supplemented with 10% fetal bovine serum (Gibco, Grand Island, NY, USA), 4 mM GlutaMAX (Gibco, Paisley, United Kingdom), 100 g/mL Streptomycin, 100 IU/mL Penicillin (SIGMA, Saint Louis USA) at 37 °C in a humidified atmosphere with 5% CO_2_. Before obtaining the sample, we treated the dishes for two hours with poly-D-lysine (diluted in Milli-Q water at 50 mg/mL). The cells were transferred to a new plate and cultured under the same conditions when they reached 80% confluence.

### 4.3. Flow Cytometry

Culture plates were trypsinized, inactivated, and supplied with DMEM/F-12 before being centrifuged at 1300 rpm for 5 min, and the supernatant was discarded. Conjugated primary antibodies were added to the pellet after it had been resuspended in PBS with 1 × 10^6^ cells per tube. Mesenchymal cells were identified using the markers CD105+ (Biolegend, San Diego, CA, USA 323218), CD73+ (BD Biosciences Franklin Lakes, NJ, USA, 563198), CD90+ (BD Biosciences, San Diego, CA USA 555597), CD45- (MACSMylteny Biotec San Diego, CA, USA 130-080-202), CD14- (BD Biosciences, San Diego, CA USA 561712), CD19- (BD Biosciences, San Diego, CA USA 560911), and CD34- (BD Biosciences San Diego, CA USA 562577). The information was recorded using a Fortessa flow cytometer (BD) and examined using the Flowjo 7.6 application.

### 4.4. Proteomic Sample

At 80% confluence, MSCs cells were grown, and proteins were removed from the culture media with three cold PBS washes. Reconstituted cells were incubated at 95 °C for 5 min in an SDT lysis solution (4% SDS (*w*/*v*), 0.1 M DTT, and 100 mM Tris/HCl pH 7.6). The samples were sonicated ten times (PEAK Bioruptor, pulse ON 30 s and OFF 30 s) at 20 °C. The supernatant was then transferred to another tube after the undisturbed cells and cell debris were separated via centrifugation at 16,000× *g* for 5 min at −20 °C. Following the manufacturer procedure, the protein concentration was assessed using a 2D Quant kit (GE Healthcare, Life SciencesPiscataway, NJ, USA).

### 4.5. Filter-Aided Sample Preparation (FASP)

FASP digestion of the protein extract was carried out according to a modified protocol described by Wiśniewski et al. [184]. We used 50 μg of protein input, and protein digestion was performed with trypsin (enzyme/protein ratio of 1: 100) overnight at 37 °C. The filtrates were transferred to new tubes and centrifuged at 14,000× *g* at 20 °C for 40 min. Subsequently, they were rinsed with 50 μL of 0.5 M NaCl and centrifuged again (twice); finally, the sample was desalted in a C18 resin column and stored at −80 °C until further use. Digestion of the samples was according to the FASP protocol described by Wiśniewski et al. [184] and modified by the LaNSE CINVESTAV proteomics team [185].

### 4.6. Liquid Chromatography-Tandem Mass Spectrometry

The resulting peptides were injected into the mass spectrometer Synapt G2-Si (Waters, Milford, MA, USA) in MS^E^ mode to calculate the area under the curve (AUC) of the Total Ion Chromatogram and thus normalize the injection in the HPLC. A Symmetry C18 Trap V/M precolumn with dimensions of 180 mm × 20 mm, a pore size of 100°, and a particle size of 5 μm was loaded precisely with tryptic peptides in each condition. The precolumn was desalted using mobile phase A, which contained 0.1% formic acid (FA) in water and mobile phase B, which had 0.1% FA in acetonitrile, under the following isocratic gradient: 99.9% mobile phase A and 0.1% of mobile phase B at a flow of 5 μL min^−1^ during 3 min. Afterward, peptides were loaded and separated on an HSS T3 C18 Column with 75 μm × 150 mm, 100 A° pore size, and 1.8 μm particle size, using a UPLC ACQUITY M-Class with the same mobile phases under the following gradient: 0 min 7% B, 121.49 min 40% B, 123.15 to 126.46 min 85% B, 129 to 130 min 7% B, at a flow of 400 nL min^−1^ and 45 °C. Data-independent acquisition technique in the HDMSE mode was used to obtain the spectra in a mass spectrometer equipped with electrospray ionization and ion mobility separation Synapt G2-Si (Waters, Milford, MA, USA). The tune page for the ionization source was set with the following parameters: 2.75 kV in the sampler capillary tube, 30 V in the sampling cone, 30 V in the source offset, 70 °C for the source temperature, 0.5 Bar for the nanoflow gas, and 150 L h^−1^ for the purge gas flow. Two chromatograms (low and high energy) were acquired in a positive mode in a range of *m*/*z* 50–2000 with a velocity of 0.5 scans s^−1^. For the high-energy chromatograms, the precursor ions were fragmented in the transfer using a collision energy ramp of 19–55 V. We analyzed the generated raw files in the DriftScope v2.8 software (Waters, Milford, MA, USA) to eliminate ions with z = 1+ and to obtain the DrifTime from each peptide detected in the mass spectrometer to generate a .rul file to calculate specific collision energy for every peptide caught in the UDMSE mode (three technical replications were carried out).

### 4.7. Data Analysis

The .raw files containing the MS and MS/MS intensities were normalized, aligned, compared, and relatively quantified using Progenesis QI for Proteomics software v3.0.3 (Waters, Milford, MA, USA) against a reversed Homo Sapiens database. The results generated from the software were exported to .xmls files to verify two levels of data quality control for label-free experiments (peptide and protein level) following the figures of merit described by Souza et al., 2017 [12] (Downloaded from UniProt, 19229 protein sequences, last modified on 16 May 2016). The parameters used for the protein identification were trypsin as a cut enzyme and one missed cleavage allowed, and carbamidomethyl (C) as a fixed modification and oxidation (M), phosphoryl (S, T, and Y) as variable modifications, in addition to automatic peptide and fragment tolerance, minimum fragment ion matches per peptide: 2, minimum fragment ion matches per protein: 5, minimum peptide matches per protein: 1, and false discovery rate ≤ 4%. Synapt G2-Si was calibrated with [Glu1]-Fibrinopeptide, [M + 2H]2+ = 785.84261 at ≤1 ppm. All the plots generated during the quality control were created using Spotfire software v7.0 (TIBCO, Palo Alto, CA). Differential expression was considered with P/C absolute ratios >1.5 and <0.66 for upregulated and downregulated proteins, respectively. The remaining proteins with ratios between 1.5 and 0.66 were considered unchanged. The ratio was calculated by dividing the average MS signal response of the three most intense tryptic peptides (Top3) of each well-characterized protein in the infected sample by the Top3 protein in the control sample. The images were created with BioRender.com.

### 4.8. Bioinformatic Analysis

For Gene Ontology (GO) terms, as well as a pathway enrichment analysis, we used WEB-based Gene SeT AnaLysis Toolkit (WEBGESTALT) “http://www.webgestalt.org/ (accessed on 03 April 2023, version 2019, Houston, TX, USA) using the Kyoto Encyclopedia of Genes and Genomes (KEGG), REACTOME https://reactome.org/ and Protein ANalysis THrough Evolutionary Relationships (PANTHER) http://pantherdb.org/ databases. The protein lists were submitted with their respective gene symbol, and a comparison was made with the existing list of the human genome. For pathways enrichment, we used the top 15 significance level. KEGG Mapper-Color https://www.genome.jp/kegg/mapper/color.html/ accessed on 21 April 2023) was used for scheme pathways.

Interactomes of differentially expressed proteins were predicted using Search Tool for the Retrieval of Interacting Genes/Proteins STRING ver. 11.5 (https://string-db.org/ accessed on 5 may 2023) with the following settings: Homo Sapiens database, text mining, experiments, database, co-expression, neighborhood, gene fusion, and co-occurrence as active interaction source with the highest confidence score (0.90).

Additionally, we used GenAnalytics (http://geneanalytics.genecards.org/ accessed on 3 april 2023) and WEBGESTALT with OMIM and DISGENET databases to find related diseases. UniProt and GeneCards (https://www.uniprot.org/ and https://www.genecards.org/ accessed on 3 april 2023) were used to identify protein characteristics and function.

## 5. Conclusions

Although many pathways, including protein misfolding and aggregation, mitochondrial dysfunction, autophagy, and oxidative stress, have been implicated in the etiology of AD, there is currently no effective treatment for the disease. Therefore, more research into disease processes and signaling pathways is required to identify new biomarkers and therapeutic targets for early diagnosis. Most pathogenic pathways have been tested in cell lines and animal models carrying FAD-causing gene mutations. Although multiple proteomics studies have recently examined dysregulated proteins in diverse biological materials collected from persons with AD, such as brain tissues and bodily fluids, relatively few studies have used patient-derived cells like fibroblasts or astrocytes. Quantitative proteome analysis is used in this study to assess the disease’s presymptomatic and symptomatic stages in olfactory MSCs derived from human carriers of the PSEN1(A431E) mutation. The MSCs have a direct connection to the illness, a high neurogenic potential, and mesenchymal properties that enable them to differentiate into several cell lineages. They are also simple to obtain and barely invasive. These fresh MSCs from patients are ideal in vitro models for upcoming research on Alzheimer’s disease.

They might also suggest that FAD patients are more vulnerable to stress due to altered energy metabolism, which might cause ROS production to rise and symptoms to manifest earlier. Our findings present a thorough proteomic analysis of MSCs from FAD patients in the early and late stages of the disease that resemble reported pathways in Sporadic Alzheimer’s disease. Although more research and validation are required to understand some altered mechanisms reported in this work, our results provide guidelines for the study and identification of new therapeutic targets and the basis for the presymptomatic diagnosis of AD.

## 6. Patents

Two patents were summited to the Mexican Institute of Intellectual Property (IMPI).

## Figures and Tables

**Figure 1 ijms-24-12606-f001:**
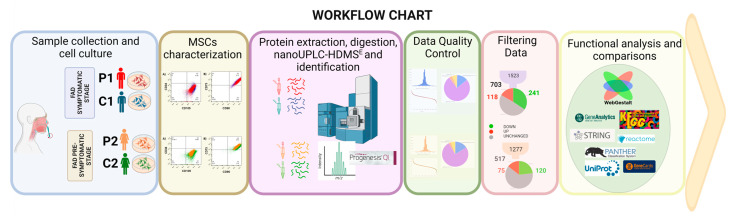
Flowchart for the study’s process. Olfactory ecto-mesenchymal stem cells were gathered and cultivated from two PSEN1 (A431E) mutant carriers and two healthy donors. MSC markers were then determined using flow cytometry. Protein was isolated and digested to identify DEPs by stage (symptomatic and presymptomatic) using two label-free proteomic techniques (P1/C1 and P2/C2). Data quality control and filtering were carried out to demonstrate proper reliability and confidence in the outcomes. Each protein group was subjected to functional analysis and comparisons using various bioinformatics techniques.

**Figure 2 ijms-24-12606-f002:**
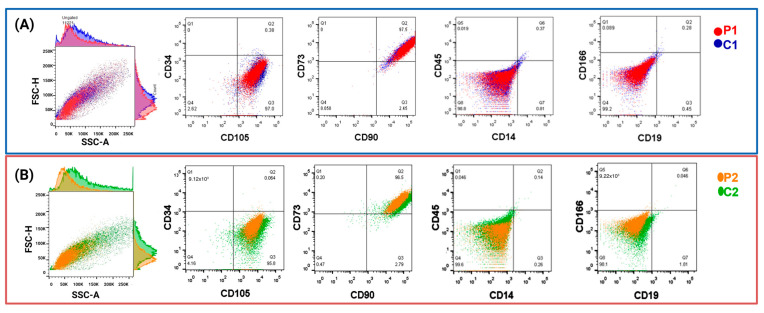
Nine passes of nasal exfoliation result in cells expressing mesenchymal markers. (**A**) The symptomatic mutation carrier (P1) showed in red. Its control (C1) is in blue, (**B**) dot plots of cells from presymptomatic mutation carrier (P2) are shown in orange, and its control (C2) is in green. Cells labeled with markers for CD34, CD105, CD73, CD90, CD45, CD14, CD166, and CD19 are shown in a cell morphology dot plot.

**Figure 3 ijms-24-12606-f003:**
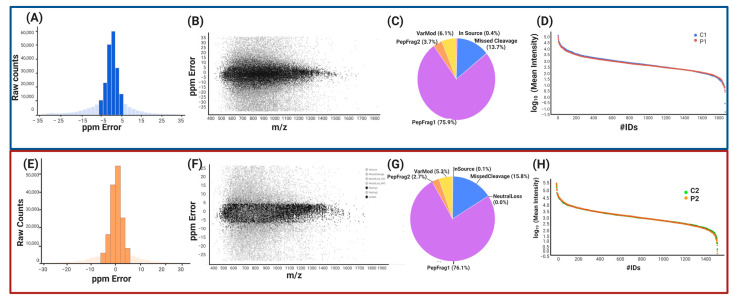
Dependability and confidence of peptides and proteins (P1/C1 and P2/C2). (**A**,**E**) histograms show the total number of peptides with an inaccuracy of no more than 5 ppm (dark blue or orange). (**B**,**F**) dot plot of PepFrag1 peptides concentrated at a maximum of 18 ppm in the case of B and 16 ppm in the case of F across the examined m/z range. (**C**,**G**) a pie graph illustrating the different types of peptides. The dynamic range of measured proteins (**D**,**H**). The Y-axis represents the average intensities for each quantified protein (expressed as log10), and the X-axis represents the number of quantified proteins (IDS).

**Figure 4 ijms-24-12606-f004:**
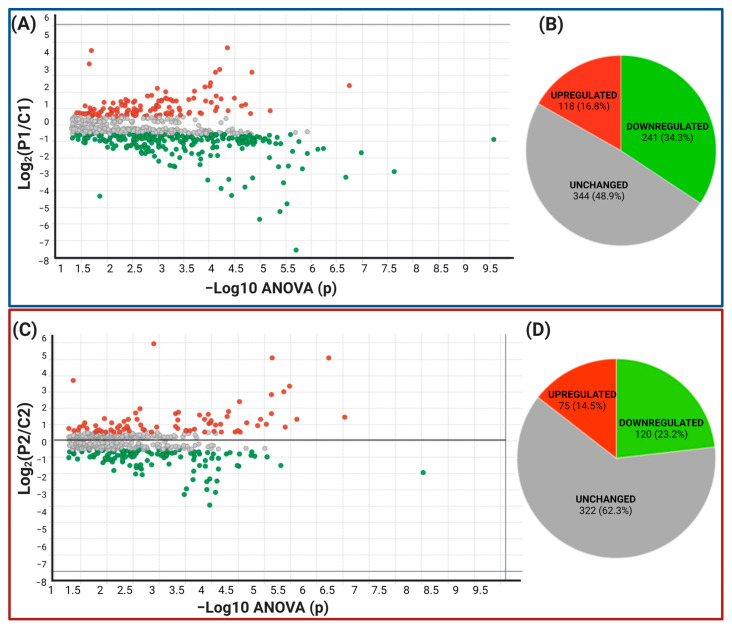
Differentially expressed proteins in symptomatic and presymptomatic carriers related to controls. Both proteins that are upregulated (red) and downregulated (green) in all pairwise comparisons are depicted in the volcano plot (**A**,**C**). (**B**,**D**) a pie chart showing the different protein classes based on differential expression.

**Figure 5 ijms-24-12606-f005:**
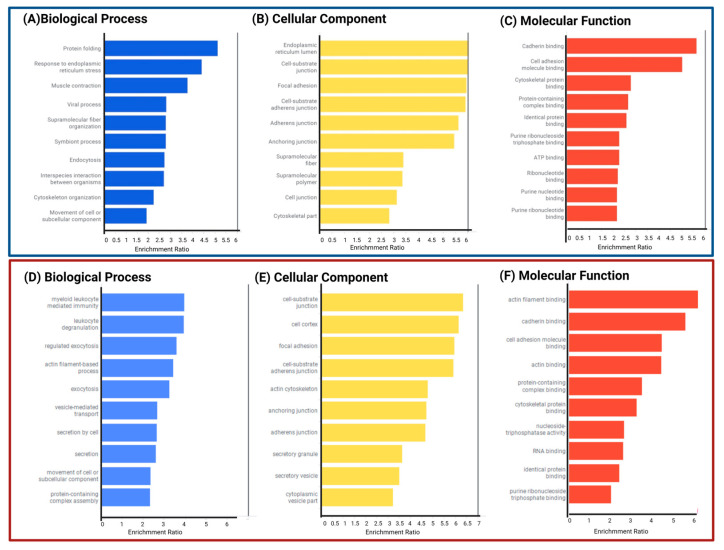
Gene ontology annotation of proteins produced differently in symptomatic (**A**–**C**) and presymptomatic (**D**,**E**) carrier cells compared to C1 and C2. Biological processes (**A**,**D**) cellular component (**B**,**E**) and molecular function (**C**,**F**).

**Figure 6 ijms-24-12606-f006:**
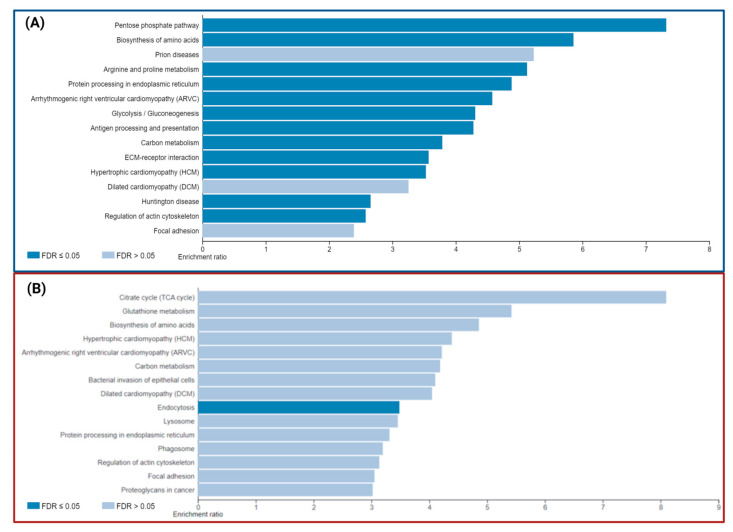
Top 15 enriched pathways in P1/C1 (**A**) and P2/C2 (**B**) using WebGestalt with KEGG Database.

**Figure 7 ijms-24-12606-f007:**
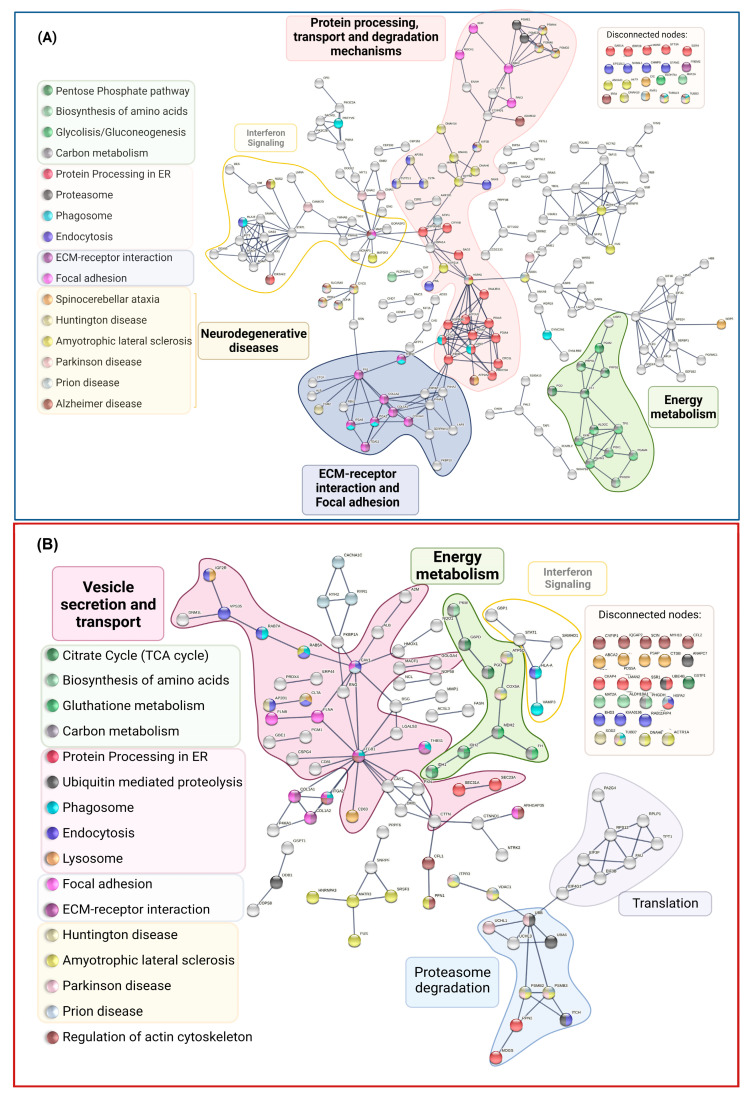
Protein–protein interaction (PPI) network of P1/C1 (**A**) and P2/C2 (**B**) DEPs constructed on STRING with the KEGG-enriched pathways underlined. The upper-right corner displays disconnected nodes connected to the indicated paths. Confidence rating (0.9).

**Figure 8 ijms-24-12606-f008:**
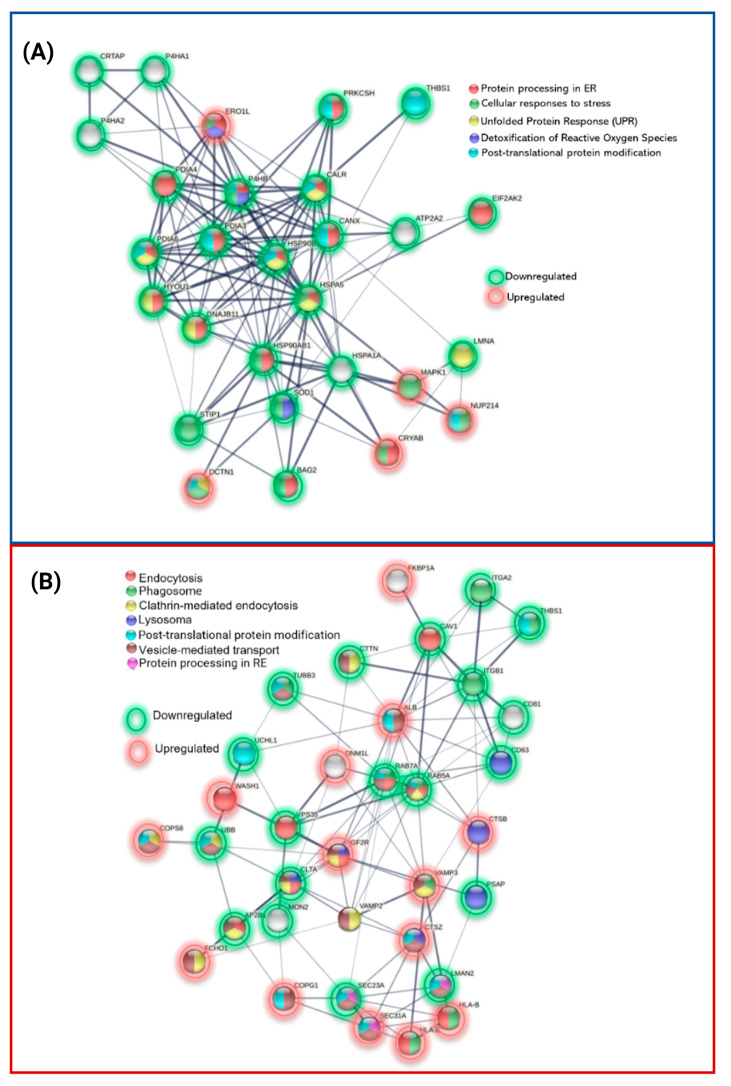
PPIs of differentially expressed proteins of P1 and P2 involved in (**A**) protein processing in ER and (**B**) vesicle secretion and transport.

**Figure 9 ijms-24-12606-f009:**
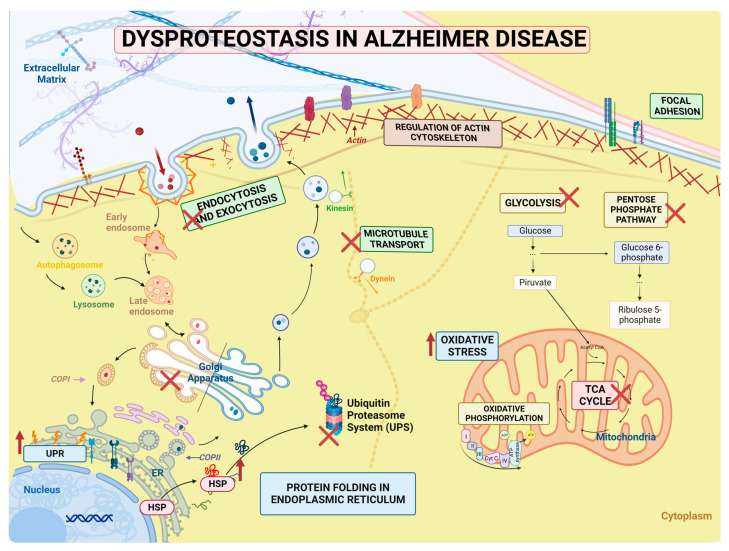
Summary of dysproteostasis network in PSEN1 (A431E) mutation carriers proteostasis comprises a complex network of processes that allow the maintenance of the conformation, concentration, and localization of proteins for their correct function. Our findings demonstrate that MSCs derived from carriers of the PSEN1 (A431E) mutation show a disruption of Protein Quality Control and proteostasis-like energy metabolism pathways that could lead to an increase in oxidative stress, an alteration in synthesis, folding, and degradation of proteins which could help to the accumulation and aggregation of misfolded proteins, and dysfunction of cytoskeleton organization, and intracellular and vesicular transport.

**Table 1 ijms-24-12606-t001:** Unique proteins for P1 and P2.

	Uniprot ID	Gene Symbol	Description	ANOVA (p)
Symptomatic carrier (P1)	O60282-2	*KIF5C*	Kinesin heavy chain isoform 5C	5.92 × 10^−9^
	Q8IV33	*KIAA0825*	Uncharacterized protein KIAA0825	7.01 × 10^−8^
	A6NCS7	*UTY*	Histone demethylase UTY	1.23 × 10^−7^
	P16401	*HIST1H1B*	Histone H1.5	3.42 × 10^−7^
	Q5JPF3-3	*ANKRD36C*	Ankyrin repeat domain-containing protein 36C	4.84 × 10^−7^
	Q9NQ36-3	*SCUBE2*	Signal peptide_ CUB and EGF-like domain-containing protein 2	8.00 × 10^−7^
	F8WBJ0	*CDK5RAP2*	CDK5 regulatory subunit-associated protein 2	1.87 × 10^−5^
	O43379-3	*WDR62*	WD repeat-containing protein 62	2.44 × 10^−5^
	Q8NEM0-3	*MCPH1*	Microcephalin	3.85 × 10^−5^
	P18859-2	*ATP5PF*	ATP synthase-coupling factor 6_ mitochondrial	1.75 × 10^−3^
	K7ESH9	*PIGN*	GPI ethanolamine phosphate transferase 1	2.50 × 10^−3^
Presymptomatic Carrier (P2)	P62633	*CNBP*	Cellular nucleic acid-binding protein	1.6801 × 10^−9^
	Q92786	*PROX1*	Prospero Homeobox protein 1	2.733 × 10^−9^
	H7BYV6	*BIN3*	Bridging integrator 3	3.9233 × 10^−9^
	G3V4C6	*RTRAF*	RNA transcription_ translation, and transport factor protein	2.3644 × 10^−7^
	O75947	*ATP5H*	ATP synthase subunit d_ mitochondrial	6.5903 × 10^−7^
	B1AKR6	*DYNLRB1*	Dynein light chain roadblock-type 1	8.1222 × 10^−7^
	Q95365	*HLA-B*	HLA class I histocompatibility antigen_ B-38 alpha chain	1.348 × 10^−5^
	A0A0C3SFZ9	*FCHO1*	F-BAR domain only protein 1	1.5595 × 10^−5^
	Q8IWT3	*CUL9*	Cullin-9	2.1129 × 10^−5^
	P54577	*YARS*	Tyrosine--tRNA ligase_ cytoplasmic	3.6024 × 10^−5^
	Q9Y5L4	*TIMM13*	Mitochondrial import inner membrane translocase subunit Tim13	0.0020643
	Q9Y2E4	*DIP2C*	Disco-interacting protein 2 homolog C	0.00959067

**Table 2 ijms-24-12606-t002:** Top 10 downregulated proteins in PSEN(A431E) mutation carriers.

	Uniprot ID	Gene Symbol (GS)	Description	FC	ANOVA (p)
Symptomatic carrier (P1)	P68871	*HBB*	Hemoglobin subunit beta	−189.5	1.99 × 10^−6^
	Q9HCM1	*KIAA1551*	Uncharacterized protein KIAA1551	−52.0	1.05 × 10^−5^
	P20591	*MX1*	Interferon-induced GTP-binding protein Mx1	−37.1	4.19 × 10^−6^
	A0A2R8Y7C0	*HBA2*	Hemoglobin subunit alpha (Fragment)	−27.6	3.04 × 10^−6^
	E7EMF1	*ITGA2*	Integrin alpha-2	−26.9	7.13 × 10^−5^
	Q86YA3	*ZGRF1*	Protein ZGRF1	−29.7	1.42 × 10^−2^
	Q8TEP8	*CEP192*	Centrosomal protein of 192 kDa	−19.2	3.72 × 10^−5^
	Q15283	*RASA2*	Ras GTPase-activating protein 2	−14.3	5.99 × 10^−5^
	Q9Y6K5	*OAS3*	2′-5′-oligoadenylate synthase 3	−13.5	2.02 × 10^−5^
	O14879	*IFIT3*	Interferon-induced protein with tetratricopeptide repeats 3	−11.2	3.56 × 10^−6^
Presymptomatic Carrier (P2)	Q9Y6U3	*SCIN*	Adseverin	−14.9	1.10 × 10^−4^
	C9J1V9	*EEF1E1-BLOC1S5*	EEF1E1-BLOC1S5 readthrough (NMD candidate)	−9.7	3.39 × 10^−4^
	Q7Z2Z1	*TICRR*	Treslin	−8.7	8.58 × 10^−5^
	P35580	*MYH10*	Myosin-10	−7.8	1.28 × 10^−4^
	Q16620	*NTRK2*	BDNF/NT-3 growth factors receptor	−7.6	3.14 × 10^−4^
	P32455	*GBP1*	Guanylate-binding protein 1	−5.6	1.28 × 10^−4^
	Q9NZR2	*LRP1B*	Low-density lipoprotein receptor-related protein 1B	−5.5	8.13 × 10^−5^
	O75131	*CPNE3*	Copine-3	−5.2	2.32 × 10^−4^
	C9JD73	*PPP1R7*	Protein phosphatase 1 regulatory subunit 7	−5.0	1.13 × 10^−4^
	A0A087X0Y2	*UTY*	Histone demethylase UTY	−4.1	2.15 × 10^−3^

**Table 3 ijms-24-12606-t003:** Top 10 upregulated proteins in symptomatic carrier (P1) compared with control (C1) MSCs.

	Uniprot ID	GS	Description	FC	ANOVA (p)
Symptomatic carrier (P1)	P25092	*GUCY2C*	Heat-stable enterotoxin receptor	34.9	2.07 × 10^−2^
	Q13523	*PRPF4B*	Serine/threonine-protein kinase PRP4 homolog	25.9	4.42 × 10^−5^
	Q15691	*MAPRE1*	Microtubule-associated protein RP/EB family member 1	19.8	2.28 × 10^−2^
	Q8TD84	*DSCAML1*	Down syndrome cell adhesion molecule-like protein 1	10.4	6.39 × 10^−5^
	Q4LDE5	*SVEP1*	Sushi_ von Willebrand factor type A_ EGF and pentraxin domain-containing protein 1	9.4	7.57 × 10^−5^
	Q96FQ6	*S100A16*	Protein S100-A16	9.3	1.45 × 10^−5^
	P11908	*PRPS2*	Ribose-phosphate pyrophosphokinase 2	5.9	9.41 × 10^−5^
	P17050	*NAGA*	Alpha-N-acetylgalactosaminidase	5.4	1.82 × 10^−7^
	O00750	*PIK3C2B*	Phosphatidylinositol 4-phosphate 3-kinase C2 domain-containing subunit beta	5.2	9.90 × 10^−5^
	A0A2R8Y5F1	*TSC2*	Tuberin	5.0	1.22 × 10^−4^
Presymptomatic Carrier (P2)	O00429	*DNM1L*	Dynamin-1-like protein	94.5	1.29 × 10^−3^
	Q99627	*COPS8*	COP9 signalosome complex subunit 8	35.0	6.07 × 10^−7^
	P07333	*CSF1R*	Macrophage colony-stimulating factor 1 receptor	34.3	7.27 × 10^−6^
	P05386	*RPLP1*	60S acidic ribosomal protein P1	19.8	4.54 × 10^−2^
	P62942	*FKBP1A*	Peptidyl-prolyl cis-trans isomerase FKBP1A	10.3	3.36 × 10^−6^
	P02768	*ALB*	Serum albumin	8.2	4.31 × 10^−6^
	J3KQ32	*OLA1*	Obg-like ATPase 1	7.3	7.48 × 10^−6^
	Q92817	*EVPL*	Envoplakin	5.3	3.00 × 10^−5^
	P50416	*CPT1A*	Carnitine O-palmitoyltransferase 1_ liver isoform	4.0	2.37 × 10^−3^
	Q9Y2X3	*NOP58*	Nucleolar protein 58	3.5	4.94 × 10^−5^

**Table 4 ijms-24-12606-t004:** KEGG pathways enrichment analysis in symptomatic carrier compared to C1.

KEGG Entry	Pathway	Size	Overlapped Proteins	Enrich Ratio	*p* Value	FDR
hsa00030	Pentose phosphate pathway	30	6 (↑: *PRPS1, PRPS2*. ↓: *ALDOC, PGD*, *PGM2, TKT*)	7.32	1.32 × 10^−4^	8.86 × 10^−3^
hsa01230	Biosynthesis of amino acids	75	12 (↑: *ALDH18A1, PRPS1, PRPS2, PHGDH*. ↓: *ALDOC, MAT2A, PGAM1, PGAM4, PGK1, PKM, TKT, TPI1*)	5.86	7.19 × 10^−7^	1.17 × 10^−4^
hsa05020	Prion diseases	35	5 (↑: *MAPK1*. ↓: *HSPA1A, HSPA5, SOD1, STIP1*)	5.23	2.41 × 10^−3^	6.04 × 10^−2^
hsa00330	Arginine and proline metabolism	50	7 (↑: *ALDH18A1, OAT*,↓: *ALDH7A1, LAP3, NOS2, P4HA1. P4HA2*)	5.13	3.77 × 10^−5^	1.76 × 10^−5^
hsa04141	Protein processing in the endoplasmic reticulum	165	22 (↑: *SAR1A, CRYAB. ERO1A*. ↓: *BAG2, CALR, CANX, DNAJB11, HSP90AB1, HSP90B1, HSPA1A, HSPA5, PDIA3, PDIA4, PDIA6, ERP29, P4HB, EIF2AK2, HYOU1, LMAN2, PRKCSH, SSR4, STT3A*)	4.88	5.29 × 10^−10^	1.73 × 10^−7^
hsa05412	Arrhythmogenic right ventricular cardiomyopathy	72	9 (↓: *ACTN2, ATP2A2, CTNNA2, DES, DSP, ITGA11, ITGA2, ITGA5, LMNA*)	4.58	1.36 × 10^−4^	8.86 × 10^−3^
hsa00010	Glycolysis/Gluconeogenesis	68	8 (↓: *ALDH7A1, ALDOC, PGAM1, PGAM4, PGK1, PGM2, PKM, TPI1*)	4.31	4.89 × 10^−4^	1.99 × 10^−2^
hsa04612	Antigen processing and presentation	77	9 (↓: *CANX, CALR, HLA-E, HSP90AB1, HSPA1A, HSPA5, PDIA3, PSME1, PSME2*)	4.28	2.29 × 10^−4^	1.24 × 10^−2^
hsa01200	Carbon metabolism	116	12 (↑: *PHGDH, PRPS1, PRPS2, SDHA,* ↓: *PKM, PGAM1, PGAM4, PGD, PGK1, TKT, ALDOC, TPI1*.)	3.79	7.08 × 10^−5^	7.70 × 10^−3^
hsa04512	ECM-receptor interaction	82	8 (↑:*COL6A3*, ↓: *COL1A2, COL1A1, FN1, ITGA11, ITGA2, ITGA5, THBS1*)	3.57	1.70× 10^−3^	5.00× 10^−2^
hsa05410	Hypertrophic cardiomyopathy (HCM)	83	8 (↓: *ATP2A2, DES, ITGA11, ITGA5, ITGA2, LMNA, TPM2, TPM3*)	3.53	1.84 × 10^−3^	5.00 × 10^−2^
hsa05414	Dilated cardiomyopathy (DCM)	90	8 (↓: *ATP2A2, DES, ITGA11, ITGA5, ITGA2, LMNA, TPM2, TPM3*)	3.25	3.07 × 10^−3^	6.68 × 10^−2^
hsa05016	Huntington disease	193	14 (↑: *SLC25A5, DCTN1, CLTCL1, CYCS, SDHA*. ↓: *AP2B1, CLTA, DNAH1, DNAH10, DNAH14, DNAH6, ITPR1, SOD1, TGM2*)	2.66	7.83 × 10^−4^	2.55 × 10^−2^
hsa04810	Regulation of actin cytoskeleton	213	15 (↑: *ENAH, MAPK1, NCKAP1, PAK2, ROCK1.* ↓: *FN1, GSN, IQGAP3, ITGA11, ITGA2, ITGA5, PAK3, PIKFYVE, RRAS, SCIN*)	2.58	6.95 × 10^−4^	2.52 × 10^−2^
hsa04510	Focal adhesion	199	13 (↑: *COL6A3, PAK2, MAPK1, ROCK1*.↓: *COL1A2, COL1A1, FN1, ITGA11, ITGA2, ITGA5, KDR, PAK3, THBS1*)	2.39	3.06 × 10^−3^	6.68 × 10^−2^

↑: upregulated protein, ↓: downregulated protein.

**Table 5 ijms-24-12606-t005:** KEGG pathways in presymptomatic carrier compared to C2.

KEGG Entry	Description	Size	Overlap	EnrRatio	*p* Value	FDR
hsa00020	Citrate cycle (TCA cycle)	30	4 (↑: *FH, MDH2*, ↓: *IDH1, IDH2*)	8.10	1.38 × 10^−3^	6.93 × 10^−2^
hsa00480	Glutathione metabolism	56	5 (↓: *G6PD, GSTP1, IDH1, IDH2,PGD*)	5.42	2.18 × 10^−3^	7.73 × 10^−2^
hsa01230	Biosynthesis of amino acids	75	6 (↓: *ALDH18A1, IDH1, IDH2, MAT2A, PHGDH, PKM*)	4.86	1.41 × 10^−3^	6.93 × 10^−2^
hsa05410	Hypertrophic cardiomyopathy (HCM)	83	6 (↓: *CACNA1C, DMD, ITGA2, ITGB1, RYR2, TPM1*)	4.39	2.37 × 10^−3^	7.73 × 10^−2^
hsa05412	Arrhythmogenic right ventricular cardiomyopathy	72	5 (↓: *CACNA1C, DMD, ITGA2, ITGB1, RYR2*)	4.22	6.48 × 10^−3^	1.51 × 10^−1^
hsa01200	Carbon metabolism	116	8 (↑: *FH, MDH2*, ↓: *G6PD, IDH1, IDH2, PGD, PHGDH, PKM*)	4.19	6.20 × 10^−4^	6.93 × 10^−2^
hsa05100	Bacterial invasion of epithelial cells	74	5 (↓: *CAV1, CLTA, CTTN, ITGB1, PXN*)	4.10	7.27 × 10^−3^	1.58 × 10^−1^
hsa05414	Dilated cardiomyopathy (DCM)	90	6 (↓: *CACNA1C, DMD, ITGA2, ITGB1, RYR2, TPM1*)	4.05	3.56 × 10^−3^	9.67 × 10^−2^
hsa04144	Endocytosis	244	14 (↑: *IGF2R, WASHC5* ↓: *AP2B1, CAV1, CLTA, EHD3, HLA-A, HSPA2, ITCH, RAB11FIP4, RAB5A, RAB7A, UBB, VPS35*)	3.48	4.38 × 10^−5^	1.43 × 10^−2^
hsa04142	Lysosome	123	7 (↑: *ABCA2, CTSB, CTSZ, IGF2R* ↓: *CD63, CLTA, PSAP*)	3.46	4.06 × 10^−3^	1.02 × 10^−1^
hsa04141	Protein processing in the endoplasmic reticulum	165	9 (↑: *SEC31A, SSR1, UBE4B* ↓: *CKAP4, HSPA2, LMAN2, MOGS, RPN2, SEC23A*)	3.31	1.54 × 10^−3^	6.93 × 10^−2^
hsa04145	Phagosome	152	8 (↑: *VAMP3*. ↓: *HLA-A, ITGA2, ITGB1, RAB5A, RAB7A, THBS1, TUBB3*)	3.20	3.50 × 10^−3^	9.67 × 10^−2^
hsa04810	Regulation of actin cytoskeleton	213	11 (↑: *ARHGAP35, CFL1, CFL2, CYFIP1, IQGAP2, PFN1,* ↓: *ITGA2, ITGB1, MYH10, PXN, SCIN*)	3.14	7.35 × 10^−4^	6.93 × 10^−2^
hsa04510	Focal adhesion	199	10 (↑: *ARHGAP35, COL1A1, COL1A2, FLNA, FLNB,* ↓: *CAV1, ITGA2, ITGB1, PXN, THBS1*)	3.05	1.58 × 10^−3^	6.93 × 10^−2^
hsa05205	Proteoglycans in cancer	201	10 (↑: *FLNA, FLNB, ↓: CAV1, CD63, CTTN, ITGA2, ITGB1, ITPR3, PXN, THBS1)*	3.02	1.70 × 10^−3^	6.93 × 10^−2^

↑: upregulated protein, ↓: downregulated protein.

**Table 6 ijms-24-12606-t006:** Summary of essential pathways altered in PSEN1(A431E) mutation carriers.

Pathways	DEPs in Symptomatic Carrier	DEPs in Presymptomatic Carrier
ENERGY METABOLISM	↑: *PRPS1* ^1,2,3^, *PRPS2* ^1,2,3^, ***ALDH18A1** *^2,6^, ***PHGDH*** ^2,3^, *SDHA* ^3,5,8^, *OAT* ^6^, *CYCS* ^8^. ↓: *ALDOC* ^1,2,3,4^, ***PGD*** ^1,3,7^, *PGM2* ^1,4^, *TKT* ^1,2,3^, ***MAT2A*** ^2^, *PGAM1* ^2,3,4^, *PGAM4* ^2,3,4^, *PGK1* ^2,3,4^, ***PKM*** **^2,3,4^**, *TPI1* ^2,3,4^, *ALDH7A1* ^4,6^, *LAP3* ^6,7^, ***P4HA1*** ^6^, *P4HA2* ^6^, *NOS2* ^6^, *GSTO1* ^7^, *PRDX6* ^7^.	↑: *FH* ^3,5^, *MDH2* ^3,5^, *GSTP1* ^7^, *COX5A* ^8^. ↓: *G6PD* ^1,3,7^, ***PGD* ^1,3,7^**, *PGM1* ^1,4^, ***ALDH18A1* ^2,6^**, *IDH1* ^2,3,5,7^, *IDH2* ^2,3,5,7^, ***MAT2A* ^2^**, ***PHGDH* ^2,3^**, ***PKM* ^2,3,4^**, ***P4HA1* ^6^**, *ATP5PO* ^8.^
VESICLE TRANSPORT AND DEGRADATION	↑: *KIF5B, STAM2 ^a^, CLTCL1 ^a,b^, NAGA ^b^, **TUBB3** ^**c**^, DYNC2H1 ^c^, TUBAL3 ^c^, PSMD2 ^d^, UBRS ^e^, MAPK1 ^f,g^,TSC2 ^g^, IRS4 ^g^, ATG4C ^g^* ↓: ***AP2B1 ^a^**^,^ **CLTA ^a,b^**, HLA-E ^a,c^, HSPA1A ^a^, KIF5C ^a^, CHMPS ^a^, PML ^a^, EPS15L1 ^a^, SNX6 ^a^, SH3GL1 ^a^, CTSD ^b,g,^, SCARB2 ^b^, PIKFYVE ^c^, **ITGA2 ^c^**, ITGA5 ^c^, **THBS1 ^c^**, CALR ^c^, CANX ^c^, PSMA4 ^d^, PSMA6 ^d^, PSME1 ^d^, PSME2 ^d^, PML ^e^, HERC1 ^e^, GSN ^f^, **SCIN** ^f^, ITPR1 ^g^, RRAS ^g^*	↑: *IGF2R ^a,b,^, WASHC5 ^a,^, ABCA2 ^b^, CTSB ^b^, CTSZ ^b^, IGF2R ^a,b^, VAMP3 ^c^, UBE4B ^e^, DDB1 ^e^, CFL1 ^f^, CFL2 ^f^, PP2CB ^g^* ↓*: **AP2B1 ^a^**, CAV1 ^a,^, **CLTA ^a,b^**, EHD3 ^a,^, HLA-A ^a,c^, HSPA2 ^a,^, ITCH ^a,e,^, RAB11FIP4 ^a^, RAB5A ^a,c^, RAB7A ^a,c,g^, UBB ^a,e^, VPS35 ^a,^, CD63 ^b^, PSAP ^b^, **ITGA2 ^c^**, ITGB1 ^c^, **THBS1 ^c^**, **TUBB3 ^c^**, PSMB2 ^d^, PSMB3 ^d^, ANAPC7 ^e^, UBA6 ^e^, MARCKS ^f^, **SCIN** ^f^*
PROTEIN PROCESSING IN THE ENDOPLASMIC RETICULUM	22 (↑: *CRYAB, SAR1A ↓: BAG2, CALR, CANX, DNAJB11, HSP90AB1, HSP90B1, HSPA1A, HSPA5, PDIA3, PDIA4, PDIA6, ERP29, ERO1A, P4HB, EIF2AK2, HYOU1, LMAN2, PRKCSH, SSR4, STT3A*)	9 (↑: *SEC31A, SSR1, UBE4B ↓: CKAP4, HSPA2, LMAN2, MOGS, RPN2, SEC23A*)
REGULATION OF ACTIN CYTOSKELETON	15 (↑:*ENAH, MAPK1, NCKAP1, PAK2, ROCK1 ↓:FN1, GSN, IQGAP3, ITGA11, **ITGA2**, ITGA5, PAK3, PIKFYVE, RRAS, **SCIN***)	11 (↑: *ARHGAP35, CFL1. CFL2, CYFIP1, IQGAP2, PFN1, ↓: **ITGA2**, ITGB1, MYH10, PXN, **SCIN***)
FOCAL ADHESION	13 (↑: *COL6A3, MAPK1, PAK2, ROCK1, ↓: **COL1A1**, **COL1A2**, FN1, ITGA11, **ITGA2**, ITGA5, KDR, PAK3, **THBS1**, FREM2)*	10 (↑: *ARHGAP35, **COL1A1**, **COL1A2**, FLNA, FLNB, ↓: CAV1, **ITGA2**, ITGB1, PXN, **THBS1***)
ARRHYTHMOGENIC CARDIOMYOPATHY	↓: *ACTN2, ATP2A2, CTNNA2, DES, DSP, ITGA11, ITGA2, ITGA5, LMNA TPM2, TPM3*	↓: *DMD, ITGA2, ITGB1, RYR2, TPM1, CACNA1C*
NEURODEGENERATIVE DISEASES (NDDs)	*↑:* ***TUBB3*** *^A,B,C,D,E^**, TUBAL3 ^A,B,C,D,E^, KIF5B ^A,B,C,D,E^, SLC25A5 ^A,B,C,D^, CYCS ^A,B,C,D,E^, PSMD2 ^A,B,C,D,E^, MAPK1 ^A^, SDHA ^A,B,C,D,E^, IRS4 ^A^, CAMK2D ^B^ **RYR1** ^C^, DCTN1 ^D,E^, CLTCL1 ^D^, NUP214 ^E^* *↓:* *ITPR1 ^A,B,C,D^, PSMA4 ^A,B,C,D,E^, PSMA6 ^A,B,C,D,E^, ADAM10 ^A^, NOS2 ^A,E^, ATP2A2 ^A^, EIF2AK2 ^A^ HSPA5 ^B^, SOD1 ^B,C,D,E^, TXN ^B^, GNAI1 ^B^, GNAI2 ^B^, HSPA5 ^C,E^, HSPA1A ^C^, STIP1 ^C^, **CLTA ^H^**, DNAH14 ^D,E^, **AP2B1 ^D^**, DNAH6 ^D,E^, DNAH10 ^D,E^, DNAH1 ^D,E^, TGM2 ^D^, U:KIF5C ^A,B,D,E^, ATP5PF ^A,B,D,E^, SETX ^E^, **FUS ^E^**, ANXA11 ^E^, MAP2K3 ^E^, **MATR3 ^E^***	↑: *VDAC1 ^A,B,C,D,E^, COX5A ^A,B,C,D,E^, MME ^A^, **RYR1** ^C^, PFN1 ^E^, SRSF3 ^E^, HNRNPA3* ^E^ ↓: ***TUBB3*** *^A,B,C,D,E^, ITPR3 ^A,B,C,E^, ATP5PO ^A,B,C,D,E^, PSMB2 ^A,B,C,D,E^, PSMB3 ^A,B,C,D,E^, CACNA1C ^A,C^, UBB ^B^, UCHL1 ^B^ CAV1 ^C^, RYR2 ^C^, HSPA2 ^C^, ACTR1A ^D,E^, **CLTA ^D^**, **AP2B1 ^D^**, DNAH6 ^D,E^, SOD2 ^D^, **FUS ^E^**, **MATR3 ^E^**, RAB5A ^E^*

↑: upregulated protein, ↓: downregulated protein, U: unique protein, **Bold proteins**: shared DEP in both mutation carriers. Pentose phosphate pathway: ^1^, biosynthesis of amino acids: ^2^, carbon metabolism: ^3^, glycolysis and gluconeogenesis: ^4^, TCA: ^5^, arginine and proline metabolism: ^6^, glutathione metabolism: ^7^, oxidative phosphorylation ^8^. Endocytosis: *^a^*, lysosome: *^b^*, phagosome: *^c^*, proteasome: *^d^*, ubiquitin-mediated proteolysis *^e^*, Fc gamma R-mediated phagocytosis *^f^*, autophagy *^g^*. Alzheimer’s disease: ^*A*^, Parkinson’s disease: ^*B*^*,* prions disease: ^*C*^*,* Huntington’s disease: ^*D*^, Amyotrophic lateral sclerosis: *^E^.*

## Data Availability

Not applicable.

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
