# Peer review of "The Proteome Profile of Olfactory Ecto-Mesenchymal Stem Cells-Derived from Patients with Familial Alzheimer’s Disease Reveals New Insights for AD Study"

_ijms, 2023, doi:10.3390/ijms241612606_

Round 1
Reviewer 1 Report
The manuscript entitled “The Proteome profile of Olfactory ecto-Mesenchymal Stem 2 Cells-derived from patients with Familial Alzheimer's Disease 3 reveals new insights for AD study” by Hernandez et al has demonstrated the proteomic differences occur in familial Alzheimer’s disease patients in olfactory ecto-mesenchymal stem cells by liquid chromatography and mass spectrometric approach. The results are interesting. Following minor changes can be incorporated in the manuscript for better readability:
11. Figure 1 can be moved to supporting information.
22. Add a detailed figure legend for Figure 10 that would help to explain the schematic.
Author Response
We truly appreciate the time and effort you dedicated to thoroughly evaluating our manuscript. We are pleased to inform you that we have carefully considered and implemented all of your suggestions into the revised version of the manuscript to improving and refining the quality of our work:
-Figure 1 was moved to supplementary figures and a detailed legend was added to Figure 10.

Reviewer 2 Report
Dear Authors
Despite the significant and interesting results obtained in the publication, I cannot accept your results because the data sample is very small. I understand that it is very difficult to find material for the work, but it does not cancel the method of statistical data.
Reviewers in one of my publications pointed out to me that electronic images of mitochondria should be from 5 animals in the control and the experiment, although it is much easier to assess the degree of change from electronic images (they are either there or not) than the data you present.
Extensive editing of English language required
Author Response
Response to Reviewer 2 Comments
Point: Despite the significant and interesting results obtained in the publication, I cannot accept your results because the data sample is very small. I understand that it is very difficult to find material for the work, but it does not cancel the method of statistical data.
Reviewers in one of my publications pointed out to me that electronic images of mitochondria should be from 5 animals in the control and the experiment, although it is much easier to assess the degree of change from electronic images (they are either there or not) than the data you present.
Response 1: Dear Reviewer, we sincerely appreciate the comments, time, and effort you dedicated to reviewing our work, but allow me to explain why our results are entirely acceptable. Hoping you can understand the importance and relevance of this work:
- According to Alzforum.org, until now, PSEN1 has 357 reported mutations, of which less than a third part are pathogenic. Most of these mutations are presented in case reports that provide phenotypic descriptions using small pedigrees.
Of all PSEN1 mutations, only three mutations have a high incidence of FAD in Latin America. The first is the E208A mutation reported in Antioquia, Colombia, with approximately 5000 carriers (Sepulveda-Falla, Glatzel, and Lopera 2012). The second population consists of eight families originating in Puerto Rico carrying the G260A (Athan et al. 2001), and finally, our study mutation in Mexico, the A431E (c.1292C>A, rs63750083) mutation with a founder effect in Jalisco. Until now, there are less than ten studies that report this mutation:
A431E mutation in PSEN1 was first identified by (Rogaeva 2002) in five patients without a report of their clinical history. Years later, (Yescas et al. 2006) described this mutation in nine unrelated Mexican families with early-onset AD and suggested that the mutation originated from a common ancestor. They reported a mean age of symptom onset of 40 years and subcortical atrophy. In the same year (Murrell et al. 2006) reported 20 patients who also carried the A431E mutation identified from 15 families, 14 of which were of Mexican mestizo descent and in 9 of which their ancestry could be traced to the state of Jalisco with similar symptoms to those reported by (Yescas et al. 2006). One article identifies low levels of Aβ1–37, Aβ1–38, and Aβ1–39 in cerebrospinal fluid in carriers of this variant compared to people with SAD (Portelius et al., 2010). (Parker et al., 2019) they have reported a homozygosity case with a most aggressive phenotype and early presentation. Until now, the group of Dr. Luis Figuera has made the most significant contribution since they have registered this mutation in approximately 75% (29/39) of the cases diagnosed with Familial Dementia of Early Onset at the Centro Médico Nacional de Occidente-IMSS Guadalajara. Although they do not specify the clinical picture of the mutation carriers, they report that it causes variability in its presentation, that some patients develop more cognitive and motor symptoms (Dumois-Petersen et al., 2020). Our research group has reported the complete clinical characterization of individuals carrying the A431E mutation(Santos-Mandujano et al., 2020). Affected subjects incorporate spastic paraparesis into their clinical picture, coinciding with other studies where high demyelination occurs(Soosman et al. 2016). Finally, recently (Orozco-Barajas et al. 2022) did a scoping review to synthesize findings related to this A431E mutation in categories such as genetics, clinical, imaging techniques, neuropsychology, neuropathology, and biomarkers. They showed a significant clinical heterogeneity reported on patients with this mutation. The evidence regarding phenotypic variability is inconclusive. Joshi et al. (2012) identified atypical characteristics in carriers of FAD-associated variants, and other authors also reported these characteristics: The pseudobulbar effect was also identified in the case of Parker et al. (2019); myoclonus occurred in one of the families identified by Yescas et al. (2006) ; gait disturbance was identified as the first symptom ( Dumois-Petersen et al., 2020 ); and headaches were reported in the case of an A431E carrier (Alakkas et al., 2020).
These papers are the only ones that have studied A431E mutation. However, no one reports this kind of result. No report uses cells from these patients, much less mesenchymal stem cells. This first proteomic label-free approach uses olfactory ecto-mesenchymal cells in AD.
References:
Athan, Eleni S., Jennifer Williamson, Alejandra Ciappa, Vincent Santana, Stavra N. Romas, Joseph H. Lee, Haydee Rondon, et al. 2001. “A Founder Mutation in Presenilin 1 Causing Early-Onset Alzheimer Disease in Unrelated Caribbean Hispanic Families.” JAMA 286 (18): 2257–63. https://doi.org/10.1001/jama.286.18.2257.
Dumois-Petersen, Sofia, Martha P. Gallegos-Arreola, María T. Magaña-Torres, Francisco J. Perea-Díaz, John M. Ringman, and Luis E. Figuera. 2020. “Autosomal Dominant Early Onset Alzheimer’s Disease in the Mexican State of Jalisco: High Frequency of the Mutation PSEN1 c.1292C>A and Phenotypic Profile of Patients.” American Journal of Medical Genetics Part C: Seminars in Medical Genetics 184 (4): 1023–29. https://doi.org/10.1002/ajmg.c.31865.
Murrell, Jill, Bernardino Ghetti, Elizabeth Cochran, Miguel Angel Macias-Islas, Luis Medina, Arousiak Varpetian, Jeffrey L. Cummings, et al. 2006. “The A431E Mutation in PSEN1 Causing Familial Alzheimer’s Disease Originating in Jalisco State, Mexico: An Additional Fifteen Families.” Neurogenetics 7 (4): 277–79. https://doi.org/10.1007/s10048-006-0053-1.
Orozco-Barajas, Maribel, Yulisa Oropeza-Ruvalcaba, Alejandro A. Canales-Aguirre, and Victor J. Sánchez-González. 2022. “PSEN1 c.1292C<A Variant and Early-Onset Alzheimer’s Disease: A Scoping Review.” Frontiers in Aging Neuroscience 14: 860529. https://doi.org/10.3389/fnagi.2022.860529.
Parker, John, Tahseen Mozaffar, Ashlynn Messmore, Joshua L. Deignan, Virginia E. Kimonis, and John M. Ringman. 2019. “Homozygosity for the A431E Mutation in PSEN1 Presenting with a Relatively Aggressive Phenotype.” Neuroscience Letters 699 (April): 195–98. https://doi.org/10.1016/j.neulet.2019.01.047.
Rogaeva, Ekaterina. 2002. “The Solved and Unsolved Mysteries of the Genetics of Early-Onset Alzheimer’s Disease.” NeuroMolecular Medicine 2 (1): 1–10. https://doi.org/10.1385/NMM:2:1:01.
Santos-Mandujano, Rosalía A., Natalie S. Ryan, Lucía Chávez-Gutiérrez, Carmen Sánchez-Torres, and Marco Antonio Meraz-Ríos. 2020. “Clinical Association of White Matter Hyperintensities Localization in a Mexican Family with Spastic Paraparesis Carrying the PSEN1 A431E Mutation.” Journal of Alzheimer’s Disease: JAD 73 (3): 1075–83. https://doi.org/10.3233/JAD-190978.
Sepulveda-Falla, Diego, Markus Glatzel, and Francisco Lopera. 2012. “Phenotypic Profile of Early-Onset Familial Alzheimer’s Disease Caused by Presenilin-1 E280A Mutation.” Journal of Alzheimer’s Disease 32 (1): 1–12. https://doi.org/10.3233/JAD-2012-120907.
Soosman, Steffan K., Nelly Joseph-Mathurin, Meredith N. Braskie, Yvette M. Bordelon, David Wharton, Maria Casado, Giovanni Coppola, et al. 2016. “Widespread White Matter and Conduction Defects in PSEN1-Related Spastic Paraparesis.” Neurobiology of Aging 47 (November): 201–9. https://doi.org/10.1016/j.neurobiolaging.2016.07.030.
Yescas, Petra, Adriana Huertas-Vazquez, María Teresa Villarreal-Molina, Astrid Rasmussen, María Teresa Tusié-Luna, Marisol López, Samuel Canizales-Quinteros, and María Elisa Alonso. 2006. “Founder Effect for the Ala431Glu Mutation of the Presenilin 1 Gene Causing Early-Onset Alzheimer’s Disease in Mexican Families.” Neurogenetics 7 (3): 195–200. https://doi.org/10.1007/s10048-006-0043-3.
Finally, the English language was edited by a native English-speaking person.

Reviewer 3 Report
The manuscript exhibits significant potential and is deserving of publication in IJMS. However, need to be improved. The most crucial concern that needs to be addressed is the lack of morphological aspects related to characterization of MSCs. Also, the expression of mesenchymal markers need to be showed in correlation with cell morphology. I suggest also involve differentiation potential, adherence, and colony formation assays.
Author Response
Response to Reviewer 3 Comments
Point: The manuscript exhibits significant potential and is deserving of publication in IJMS. However, need to be improved. The most crucial concern that needs to be addressed is the lack of morphological aspects related to the characterization of MSCs. Also, the expression of mesenchymal markers need to be showed in correlation with cell morphology. I suggest also involving differentiation potential, adherence, and colony formation assays.
Response:
Dear reviewer, thanks for your valuable comments. Your feedback is highly valued, and we appreciate your ongoing support and engagement in the review process.
We have carefully considered and implemented your suggestions into the revised version of the manuscript. First, a detailed paragraph with references was added to the introduction to clarify some of the properties of these cells (including differentiation, adherence, and clonogenicity) since many groups have already studied and characterized them. They even have been used for regenerative therapy studies. We followed the isolation method of Benítez King and Cols (Benítez-King et al. 2011), which team has already made a deep characterization of these cells:
A potential source of neural stem cells is the olfactory mucosa, where neurogenesis is necessary to replace the olfactory neurons. Olfactory ecto-mesenchymal stem cells (MSCs) have recently been discovered (Benítez-King et al. 2011; Jiménez-Acosta et al. 2023; Duan and Lu 2015) and possess unique properties compared to mesenchymal stem cells isolated from other tissue sources. First, these cells show a self-renewal and a clonal and neurosphere formation capacity(Delorme et al. 2010; Jiménez-Vaca et al. 2018; Duan and Lu 2015). Due to their ectoderm origin, they have a higher neurogenic potential than mesodermal lineage. However, they have been differentiated into non-neural lineage in vitro and in vivo(Murrell et al. 2005; 2009; Ge et al. 2015; Veron et al. 2018; Delorme et al. 2010), thus have been proposed for regenerative therapy for multiple diseases (Murrell et al. 2008; McDonald et al. 2010; Stamegna et al. 2018; Riquelme et al. 2020). These cells have been obtained in postmortem tissues and by invasive methods such as mucosal and epithelial biopsies. They can be obtained from the olfactory nasal niche using non-invasive techniques (Benítez-King et al. 2011). Additionally, these cells are crucial to the disease because most AD patients also have hyposmia years before symptoms appear (Dan et al. 2021; Kotecha et al. 2018).
(Delorme et al. 2010), they reported that the olfactory cells exhibited a robust inclination to differentiate into osteoblasts. However, it failed to produce chondrocytes and gave rise to a restricted number of adipocytes. However, (Murrell et al. 2009) show that they can be differentiated into chondrogenic phenotype and other tissues like cardiac, liver, muscle, and brain(Murrell et al. 2005).
(Ge et al. 2015) characterized human olfactory mucosa mesenchymal stem cells (hOM-MSCs) and reported that these cells expressed CD73 and CD90 but no CD34 and CD45 and could be differentiated into adipocyte, osteocyte, neural stem cells, and neural cells.
In addition, another study by (Veron et al. 2018) isolated and characterized the olfactory ecto-mesenchymal stem cells from eight mammalian genera (mouse, rat, rabbit, sheep, dog, horse, gray mouse lemur, and macaque) and reported that most of them exhibit as in humans a fibroblastic morphology, an ability to form spheres, high clonal efficiency, and proliferation rate, a robust expression of nestin, a similar expression of surface markers (CD44, CD73) and the ability to differentiate into mesodermal lineages (osteoblasts, chondroblasts and tenoblasts).
Regarding placing markers concerning morphology, we added a dot plot with FSC and SCC in Figure 2. We also included a supplementary image featuring histograms for each marker and sample. We decided to present the results in this manner as it allows for a clear comparison of both comparable populations and reduces the complexity of the figure. However, we remain open to any suggestions or comments you may have regarding this approach.

Round 2
Reviewer 2 Report
Dear authors thank you for the work you have done both in conducting the research and responding to the comment.
The work is done quite neatly. Good luck!
Minor editing of English language required.
Author Response
Dear reviewer, thank you for your valuable time and work in carefully examining our manuscript. We are happy to know that, after your suggestions, the enhancement and improvement of the quality of our work is notable.

Reviewer 3 Report
The manuscript has been supplemented with additional data characterizing cells at the flow cytometer level. I accept revisions and response proposed by the authors; however, I suggest including cell morphology in the supplementary materials.
Author Response
We appreciate the reviewer's time and effort in reading through our manuscript carefully. We are pleased to inform you that all of your suggestions were carefully considered and implemented into the revised version of the manuscript to strengthen and improve the quality of our work:
We add Figure S2 with photographs of the morphology of the cell in supplementary figures.
